# Enhancing Text-to-Image Diffusion Transformer via Split-Text Conditioning

Yu Zhang[1]   Jialei Zhou[1]   Xinchen Li[1]   Qi Zhang[1]   Zhongwei Wan[2]
Duoqian Miao[1]*   Changwei Wang[1]   Longbing Cao[3]

[1]Tongji University   [2]The Ohio State University   [3]Macquarie University

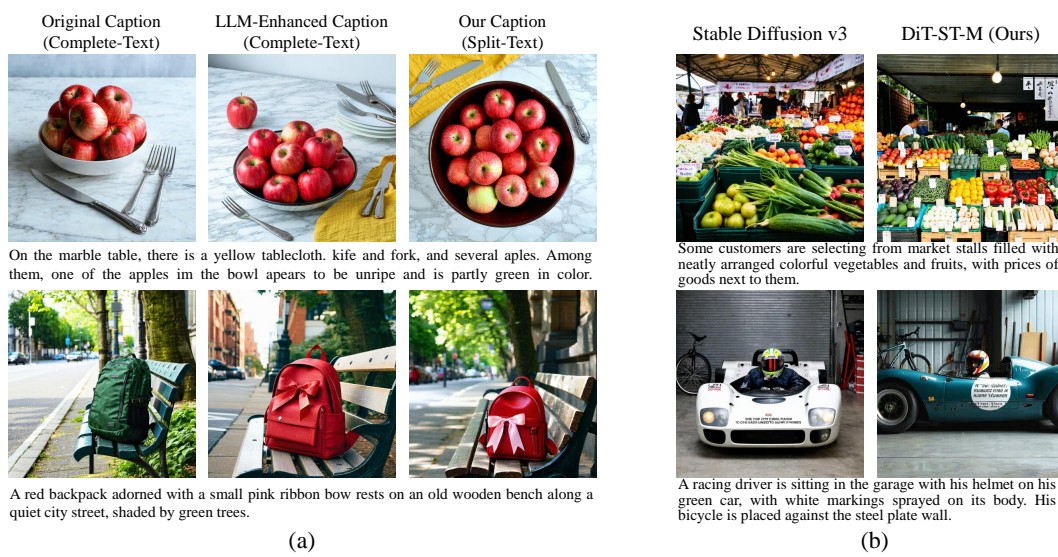

| Original Caption (Complete-Text) | LLM-Enhanced Caption (Complete-Text) | Our Caption (Split-Text) | | Stable Diffusion v3 | DiT-ST-M (Ours) |

On the marble table, there is a yellow tablecloth, kife and fork, and several aples. Among them, one of the apples im the bowl apears to be unripe and is partly green in color.

Some customers are selecting from market stalls filled with neatly arranged colorful vegetables and fruits, with prices of goods next to them.

A red backpack adorned with a small pink ribbon bow rests on an old wooden bench along a quiet city street, shaded by green trees.

A racing driver is sitting in the garage with his helmet on his green car, with white markings sprayed on its body. His bicycle is placed against the steel plate wall.

(a)                                                          (b)

Figure 1: (a) Images generated by MM-DiT 8B-E using different forms of the same caption. Our split-text caption enables the model to notice details, such as ***unripe and partly green***, ***pink ribbon bow***, and display them in the generated image. (b) Comparison of text-to-image generation results between Stable Diffusion v3 and ours. Obviously, ours has better semantic details and mitigates the attribute misbinding such as ***green car***.

## Abstract

Current text-to-image diffusion generation typically employs complete-text conditioning. Due to the intricate syntax, diffusion transformers (DiTs) inherently suffer from a comprehension defect of complete-text captions. One-fly complete-text input either overlooks critical semantic details or causes semantic confusion by simultaneously modeling diverse semantic primitive types. To mitigate this defect of DiTs, we propose a novel split-text conditioning framework named DiT-ST. This framework converts a complete-text caption into a split-text caption, a collection of simplified sentences, to explicitly express various semantic primitives and their interconnections. The split-text caption is then injected into different denoising stages of DiT-ST in a hierarchical and incremental manner. Specifically, DiT-ST leverages Large Language Models to parse captions, extracting diverse primitives and hierarchically sorting out and constructing these primitives into a split-text input. Moreover, we partition the diffusion denoising process according to its differential sensitivities to diverse semantic primitive types and determine the appropriate

---

*Corresponding Author

39th Conference on Neural Information Processing Systems (NeurIPS 2025).

timesteps to incrementally inject tokens of diverse semantic primitive types into input tokens via cross-attention. In this way, DiT-ST enhances the representation learning of specific semantic primitive types across different stages. Extensive experiments validate the effectiveness of our proposed DiT-ST in mitigating the complete-text comprehension defect. Datasets and models are available.

# 1  Introduction

In recent years, diffusion models [1, 2] have achieved unprecedented breakthroughs [3, 4, 5, 6]. Leveraging the Transformer [7] backbone, the Diffusion Transformer (DiT) [8] has rapidly become the mainstream paradigm for text-to-image generation and achieves impressive performance.

Current DiTs for text-to-image generation mostly employ complete-text captions as conditioning, which are sourced from human-curated datasets or generated by large language models (LLMs) or multimodal large language models (MLLMs). Such complete-text captions, particularly long ones, typically involve complex syntax with intertwined semantic primitives (object, relation, and attribute) and potential redundancy. When dealing with this form of captions, DiTs exhibit an inherent **complete-text comprehension defect**, reflecting in issues such as attribute misbinding [9, 10], style dominance [11, 12], semantic blending [13, 14], and semantic entanglement [15]. The comprehension defect stems from two key aspects: *insufficient semantic analysis* and *premature information exposure*.

On the one hand, *insufficient semantic analysis* arises from multiple factors. For instance, (*i*) text length bottleneck [16]: CLIP text encoder [17] accepts at most 77 tokens, yet effective tokens seldom exceed 20, making it difficult to process detailed text and leading to the truncation or loss of tail information in long captions; (*ii*) softmax competition [18, 19]: tokens compete against each other in the softmax function, causing the representation capability to decrease as token number increases; (*iii*) positional bias [20]: CLIP prioritizes the first item, making it difficult to attend to later items and leading to semantic distortion. These factors collectively hinder models from accurately analyzing semantic primitives and highlighting important semantic information from long complete-text captions. Consequently, models tend to overlook and omit crucial semantic information within a complete-text caption. However, a split-text caption derived by hierarchically sorting out the semantic primitives within the complete-text caption may reduce the syntactical complexity and comprehension difficulty for models, helping improve semantic analysis.

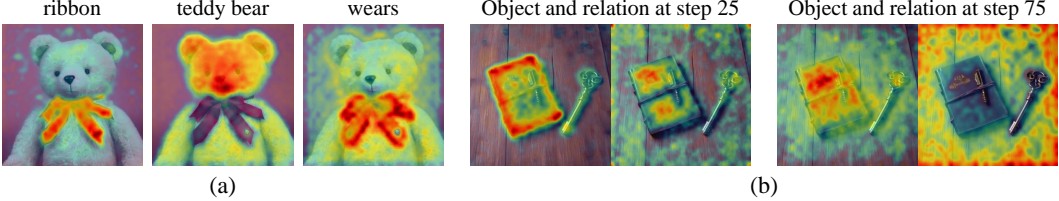

|ribbon|teddy bear|wears|Object and relation at step 25|Object and relation at step 75|
(a)                                              (b)

Figure 2: (a) Attention maps [15] for various semantic primitives. Caption: A teddy bear wearing a red ribbon around its neck. Attentions exhibit significant overlap between the object primitive 'ribbon' and relation primitive 'wears', resulting in semantic entanglement. (b) Superimposed attention maps of object primitive type and relation primitive type at denoising timesteps 25 and 75, respectively. Notably, the model focuses more on object primitives during the earlier stage and shifts more attention to relation primitives in the later stage.

On the other hand, the one-time input of a complete-text caption leads to various semantic primitives being modeled simultaneously. Since these primitives cannot be disentangled effectively within the shared representation space, ultimately competing for representation within the same image regions. This is the main reason for phenomena such as semantic entanglement, as illustrated in Figure 2 (a). We summarize this problem as *premature information exposure*, where excessive fine-grained details are prematurely exposed to the model before the model establishes stable primary semantic concepts. Existing studies [21, 22, 23, 24] indicate that diffusion models establish primary semantic concepts during the early denoising stage, while improving fine-grained details in the later stage. Details mainly originate from attribute primitives, while primary semantic concepts derive from object and relation primitives. In addition, as illustrated in Figure 2 (b), different denoising stages have diverse sensitivities of semantic primitive types. We can conclude that the prioritization order for semantic primitive types is object-relation-attribute. Therefore, we propose incrementally injecting diverse types of semantic primitives by prioritization order to improve the proportion of sensitive

primitive types at each stage, enabling the model to enhance the representation learning of a specific sensitive primitive type at the corresponding stage. As discussed above, since a split-text caption is derived by hierarchically sorting out the semantic primitives and possesses a distributed structure, it can effectively group the same type of semantic primitives together and be injected incrementally, naturally helping mitigate premature information exposure.

In this paper, we propose a novel framework, DiT-ST, to enhance the text-to-image DiT via split-text caption. A split-text caption is essentially a hierarchical collection of simplified sentences that explicitly express various semantic primitives and their interconnections, effectively reducing the syntactical complexity to alleviate insufficient semantic analysis and grouping the same type semantic primitives to be injected incrementally to address premature information exposure. Briefly, we first employ LLM [25] for caption parsing to extract various semantic primitives and assemble a hierarchical caption parsing graph. We then construct the hierarchical collection of simplified sentences according to the graph to realize the conversion from complete-text caption to split-text caption. We refine the encoding of the MM-DiT backbone [26], resulting in a DiT-ST with richer semantic level. In addition, we determine appropriate injection timesteps for three semantic primitive types by analyzing cross-attention convergence phenomena [22] and identifying the inflection point in the signal-to-noise ratio curve during denoising. After that, incrementally injecting object-relation-attribute tokens to input tokens via cross-attention to increase the proportions of corresponding sensitive semantic primitive types, further enhancing representation learning of specific semantic primitives at different stages.

Extensive experiments confirm the benefits of our split-text captioning. On GenEval, our DiT-ST-M achieves 0.69 overall accuracy and 11.3% gain over SDv3 Medium. On COCO-5K, it records the highest average CLIPScore (34.09) and the lowest FID (22.11), delivering performance competitive with SDv3.5 Large. All results and visualizations demonstrate that the method is both parameter-efficient and architecture-agnostic, yielding finer detail and stronger semantic fidelity.

## 2 Related Work

### 2.1 Diffusion Transformers

Diffusion models [2, 4, 27] reverse a fixed noising process by imitating the denoising trajectory through learned optimization. When integrated with multimodal learning [28, 29, 30, 31, 32] and other techniques [33, 34, 35, 36, 37, 38], they offer a powerful generation framework for visual content [39, 40, 41, 42, 43] and other downstream tasks [44, 45, 46, 47, 48]. Recent advances, Diffusion Transformer [8] replaces U-Net [49] with Transformer [7], enabling its strong capability for long-range dependency modeling, scalability, and flexibility. DiT models, exemplified by DeepFloyd-IF [3], Flux [50], PixArt-$\alpha$ [51] and DreamEngine [52], have successfully achieved remarkable generation performance, making DiT the mainstream paradigm for text-to-image generation.

### 2.2 Complete-Text Comprehension Defect

Complete-text comprehension defect refers to models' difficulty in effectively analyzing and understanding complete text, stemming from inherent encoder limitations or structural deficiencies, such as text length bottleneck [16], softmax competition [18, 19], positional bias [20], incorrect tokenization [53], and frequency bias [54]. This defect results in suboptimal generation quality and generated content that struggles to faithfully correspond to the prompt. Manifestations of complete-text comprehension defect include attribute misbinding [55], object missing [56], style dominance [11], semantic blending [13], semantic entanglement [15], and etc.

Many existing methods have mitigated specific manifestations. Attend-and-Excite [9] activates dominant token attention to guide the generation process in adherence to the textual prompt. WiCLP [57] and research [10] address attribute misbinding and object omission by refining the text embedding space and reweighting token-level attention to enhance compositional fidelity during generation. DeaDiff [12] mitigates style dominance by employing exclusive subsets of cross-attention layers for disentangling style and semantics. LongAlign [58] enhances semantic coherence by introducing explicit linguistic structures. In contrast, SCoPE [59] operates at the sub-prompt level, beginning with the simplest sub-prompt, then progressively introducing more complex variants and determining injection timesteps using proportional similarity ratios to mitigate alignment degradation.

While these methods indeed mitigate the complete-text comprehension defect from various perspectives, they primarily optimize for specific manifestations. Instead of addressing individual encoder limitations, we target the fundamental cause of the comprehension defect—the inherent inability of models to handle complex syntax within complete-text captions. Convert the form of captions to pursue a comprehensive mitigation strategy for the complete-text comprehension defect.

## 3  Methodology

The overall framework of DiT-ST is illustrated in Figure 3, which incorporates three key components: Caption Parsing, Hierarchical Caption Input, and Incremental Primitive Injection. The original caption $\mathcal{C}_{CT}$ adopts a complete-text structure. To sort out and refine semantic primitives, DiT-ST employs LLM ✦ as for caption parsing, extracting various semantic primitives and assembling them into a caption parsing graph $G$. According to the relationships among various semantic primitives in graph $G$, we construct them into a hierarchical collection of simplified sentences—the split-text caption $\mathcal{C}_{ST}$—which is subsequently input to the text-to-image DiT (MM-DiT). During the DiT denoising process, we calculate and determine appropriate timesteps for injecting each of the three semantic primitive types. Following the object-relation-attribute order, we incrementally inject encoded semantic primitive tokens into input tokens via cross-attention thereby enhancing the representation learning of specific semantic primitives across different stages.

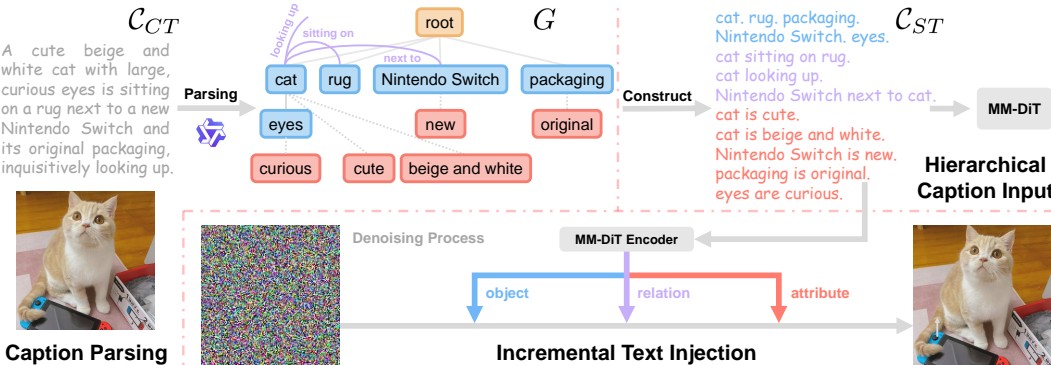

Figure 3: The overall framework of DiT-ST. Three colors represent object, relation and attribute, respectively.

### 3.1  Caption Parsing

Caption parsing aims to split the complete-text structure of captions, resulting in a caption parsing graph $G$. The reason why we choose the graph as the target product is that a hierarchical tree-like structure [60] can clearly represent various semantic primitives and their interconnections, naturally resolving existing problems of DiTs in this aspect. Therefore, we leverage a LLM (specifically Qwen-Plus ✦ [61] in this work) as a semantic analyzer and graph generator to extract various semantic primitives and assemble these semantic primitives to obtain the caption parsing graph $G$.

We define a caption parsing graph as $G = (V, E)$, where $V$ denotes the set of $N$ nodes and $E$ denotes the set of edges. Specifically, $V = \{v_i \mid i = 1, 2, \ldots, N\}$, Each node $v_i$ is defined as $v_i = (o_i, \mathcal{A}_i)$, with $o_i$ denoting the $i$-th object primitive and $\mathcal{A}_i = \{a_j\}$ representing the set of attributes associated with object primitive $o_i$. $\mathcal{O}$ is the object set, and $\mathcal{A}$ is the attribute set. Relation set $\mathcal{R}$ consists of $M$ semantic relationships: $\mathcal{R} = \{r_k \mid k = 1, 2, \ldots, M\}$. Each edge $e$ is modeled as a labelled triple $e = (v_i, v_j, r_k) \in V \times V \times \mathcal{R}$, the set of edges $E \subseteq V \times V \times \mathcal{R}$.

During the graph assembly, we use the root node to represent the entire caption, and recursively decompose it into various object primitive nodes. Relations between object primitives are represented by edges between nodes. Attributes are represented as child nodes attached to their corresponding object primitive nodes. Therefore, for a given complete-text caption $\mathcal{C}_{CT}$, the process to obtain the caption parsing graph $G$ can be formalized as follows:

$$\mathcal{C}_{CT} \xrightarrow{\text{LLM Parsing}} (\mathcal{O}, \mathcal{R}, \mathcal{A}) \xrightarrow{\text{Graph Assembly}} G. \tag{1}$$

In this way, we convert a complete-text caption into a split and hierarchical caption parsing graph.

## 3.2 Hierarchical Caption Input

The purpose of hierarchical caption input is to construct a split-text caption based on its caption parsing graph. In addition, in order to make the model better adapt to the split-text caption and enrich the overall semantic level of the input, we also modify the existing text encoding method in text-to-image DiT to better encode the split-text caption. Thus, the hierarchical caption input process consists of two components: split-text caption construction and DiT text encoding refinement.

### 3.2.1 Split-Text Caption Construction

The hierarchical structure of the caption parsing graph facilitates us to construct a split-text caption, which is a hierarchical collection of simplified sentences. Considering that inherent positional bias in text encoders (e.g., CLIP) is unavoidable during subsequent split-text encoding, we also take into account the order of semantic primitives, i.e., position the important semantic primitives at the beginning to achieve better performance. Specifically, we first rerank object primitives in the object set $\mathcal{O}$ in descending order based on the node degree and the frequency of occurrence in the caption, obtaining a reranked object set $\mathcal{O}'$. We then rerank the primitives in the relation set $\mathcal{R}$ and attribute set $\mathcal{A}$ to ensure that relations or attributes involving object primitives maintain consistency with the object primitive order in $\mathcal{O}'$, resulting in $\mathcal{R}'$ and $\mathcal{A}'$.

Subsequently, based on the reranked sets $\mathcal{O}', \mathcal{R}', \mathcal{A}'$ and following the object-relation-attribute hierarchical order, we generate corresponding simplified sentences for each primitive within these sets. The forms of simplified sentences corresponding to objects, relations, and attributes present respectively [OBJECT] object_$i$ , [RELATION] object_$i$ relation_$k$ object_$j$ , and [ATTRIBUTE] object_$i$ is attribute_$i$ . The collection comprising all these simplified sentences constitutes the split-text caption $\mathcal{C}_{ST}$. The process to construct the split-text caption can be formalized as follows:

$$(\mathcal{O}, \mathcal{R}, \mathcal{A}) \xrightarrow{\text{Acc.to } G \text{ to Rerank}} (\mathcal{O}', \mathcal{R}', \mathcal{A}') \xrightarrow{\text{Construct}} \mathcal{C}_{ST}. \qquad (2)$$

### 3.2.2 DiT Text Encoding Refinement

We employ the current mainstream multimodal diffusion transformer, MM-DiT [26], for text-to-image generation. MM-DiT adopts three text encoders: CLIP-L/14, CLIP-G/14, and T5 XXL. For a split-text caption $\mathcal{C}_{ST}$, after three encoders' text encoding, three token sequences can be obtained: $T_{L/14}^{ST} \in \mathbb{R}^{L \times D_{L/14}}, T_{G/14}^{ST} \in \mathbb{R}^{L \times D_{G/14}}$ and $T_{T5}^{ST} \in \mathbb{R}^{L \times D}$, where $L$ is the token sequence length ($L \leq 77$), $D_{L/14}, D_{G/14}$ and $D$ are the dimensions of the three token sequences.

According to the original design, the concatenation of sequences $T_{L/14}^{ST}$ and $T_{G/14}^{ST}$ yields a new token sequence whose still dimension remains smaller than $D$. Given that the new token sequence must be appended with $T_{T5}^{ST}$ for input into the MM-DiT blocks, the dimension capacity remains underutilized. Therefore, we consider fully utilizing this underutilized dimension capacity by incorporating the complete-text caption $\mathcal{C}_{CT}$ to enrich the overall semantic level of the input. As shown in Figure 4, We supplement

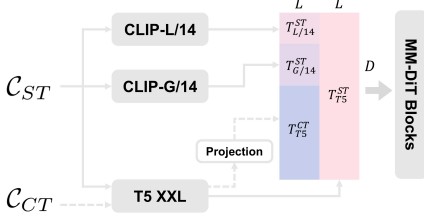

Figure 4: DiT text encoding refinement.

it by adding $\mathcal{C}_{CT}$ encoded through T5 XXL and followed by a linear projection, generating the token sequence $T_{T5}^{CT} \in \mathbb{R}^{L \times D'}$, where $D' = D - D_{L/14} - D_{G/14}$. We perform dimensional concatenation and sequence append as follows:

$$T_{concate} = Concate\left(T_{L/14}^{ST}, T_{G/14}^{ST}, T_{T5}^{CT}\right) \in \mathbb{R}^{L \times D}, \qquad (3)$$

$$T = Append\left(T_{concate}, T_{T5}^{ST}\right) \in \mathbb{R}^{2L \times D}. \qquad (4)$$

Through the above operations, we ultimately construct the input token sequence $T$ which possesses a hierarchical structure of semantic primitives and rich semantic representation.

## 3.3 Incremental Primitive Injection

As previously discussed, due to different denoising stages have differential sensitivities of semantic primitive types, prematurely exposing fine-grained detail information before the model has established

stable primary semantic concepts is detrimental to generation. We propose incrementally injecting diverse semantic primitive types to enhance differential representation learning for diverse semantic primitive types at different stages. Therefore, incremental primitive injection involves both injection timestep selection and semantic primitive injection.

### 3.3.1 Injection Timestep Selection

Since diffusion models' prioritization order for diverse semantic primitive types during the denoising process generally follows object-relation-attribute, we should, ideally, identify accurate timesteps for injecting semantic primitives. However, it is practically impossible for several reasons. First, semantic emergence occurs gradually rather than in discrete stages, lacking clear natural boundaries. Second, there are differences across samples, and the semantic emergence process can shift or stretch on the timeline with changes in prompts, resolution, and random seeds. Furthermore, adaptively selecting specific timestep for each sample will incur substantial computational costs. Therefore, we consider statistical rules across samples to identify approximately appropriate injection timesteps.

**Determine the injection timestep for attribute primitives.** Research [22] indicates that cross-attention outputs converge to a fixed point after several inference steps. This point divides the denoising process into two stages: semantic-planning and fidelity-improving. Since fidelity-improving stage primarily relies on attribute information, we select the timestep corresponding to this convergence point as the injection timestep for attribute primitives. Considering statistical rules across a batch of samples, for a sample, we first define the attention weights output by the $h$-th attention head in the $m$-th cross-attention layer at inference step $t$ as $Attn_t^{(m,h)} \in \mathbb{R}^{Q_m \times K_m}$, where $Q_m$ and $K_m$ represent the number of query tokens and key tokens, respectively. Subsequently, we calculate the difference of single-head attention between adjacent inference steps using the Frobenius norm:

$$\Delta_t^{(m,h)} = \frac{||Attn_t^{(m,h)} - Attn_{t-1}^{(m,h)}||_F}{||Attn_{t-1}^{(m,h)}||_F + \theta}, \tag{5}$$

where $\theta$ serves as a numerical stabilizer, typically set to $10^{-8}$. Then, we average across all $H$ attention heads and all $M$ cross-attention layers to compute the cross-attention difference for the sample:

$$\Delta_t^{(sample)} = \frac{1}{M} \frac{1}{H} \sum_{m=1}^{M} \sum_{h=1}^{H} \Delta_t^{(m,h)}. \tag{6}$$

By calculating the average cross-attention across all samples in the batch, we obtain the average cross-attention difference $\bar{\Delta}_t$ at $t$ inference step. We employ the moving average method to calculate the average cross-attention $\widehat{\Delta}_t$ across multiple inference steps and set a threshold $\tau$ to identify the inference step $t^*$ at which cross-attention converges:

$$\widehat{\Delta}_t = \frac{1}{w} \sum_{k=0}^{w-1} \bar{\Delta}_{t-k}, \qquad t^* = \min\{t : \widehat{\Delta}_t < \tau\}, \tag{7}$$

where $w$ denotes the size of the moving window. Thus, the timestep corresponding to inference step $t^*$ is selected as the injection timestep $s_{attr}$ for attribute primitives.

**Determine the injection timestep for relation primitives.** As previously discussed, given the entire denoising process consists of $S$ timesteps, the first $S - s_{attr}$ timesteps constitute the semantic-planning stage, during which the samples maintain a relatively high signal-to-noise ratio (SNR). Research [21] indicates that semantic concepts are primarily established at a high SNR condition. By analyzing the variation of SNR across timesteps, we observe an initial rapid decline followed by a gradual decrease, as shown in Figure 5. Considering the prioritization order for semantic primitives as object-relation-attribute, intuitively speaking, the inflection point of the SNR within the first $S - s_{attr}$ timesteps can serve as an appropriate injection timestep for relation primitives.

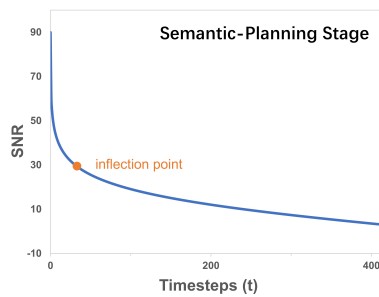

Figure 5: Inflation point of SNR.

We calculate the average SNR $\overline{snr}_t$ at each timestep $t$ across all samples in the batch for the first $S - s_{attr}$ timesteps, and construct a discrete curve function $y_t = g(\overline{snr}_t)$, where $t = 0, 1, \ldots, S - s_{attr} - 1$. We employ the discrete maximum-curvature method to identify the inflection point. First, we calculate the first-order and second-order finite differences as follows:

$$\Delta y_t = \frac{y_{t+1} - y_{t-1}}{\overline{snr}_{t+1} - \overline{snr}_{t-1}}, \qquad \Delta^2 y_t = \frac{y_{t+1} - 2y_t + y_{t-1}}{\left(\frac{1}{2}\left(\overline{snr}_{t+1} - \overline{snr}_{t-1}\right)\right)^2}. \tag{8}$$

Then, we calculate the discrete curvature:

$$\kappa_t = \frac{\left|\Delta^2 y_t\right|}{\left(1 + (\Delta y_t)^2\right)^{3/2}}. \tag{9}$$

In this way, we can obtain the index of inflection point $t^\# = \arg\max \kappa_t$, which also is the injection timestep $s_{rel}$ for relation primitives.

**Determine the injection timestep for object primitives.** Considering the limited number of timesteps before $s_{rel}$, further partition of these timesteps lacks theoretical and computational evidence. For simplicity, we set the timestep at $t = 0$ as the injection timestep $s_{obj}$ for object primitives.

By identifying the convergence point of cross-attention and the inflection point in SNR curve during denoising, we are able to adaptively determine the appropriate injection timesteps according to statistical rules and characteristics of the sample set.

### 3.3.2 Semantic Primitive Injection

Semantic primitive enhancement aims to increase the proportion of corresponding semantic primitives to align the semantic primitive type sensitivity of each denoising stage. We employ the cross-attention mechanism and incrementally inject each primitive type $\mathcal{C}^{prim}$ of the split-text caption $\mathcal{C}^{ST}$. $\mathcal{C}^{prim}$ is also encoded by three text encoders (CLIP-L/14, CLIP-G/14, and T5 XXL), obtaining $T_{L/14}^{prim} \in \mathbb{R}^{L \times D_{L/14}}$, $T_{G/14}^{prim} \in \mathbb{R}^{L \times D_{G/14}}$, and $T_{T5}^{prim} \in \mathbb{R}^{L \times D'}$, where $D_{L/14} + D_{G/14} + D' = D$. We then concate these token sequences along the dimension to obtain the primitive injection token sequence $T^{prim}$, which is used for cross-attention calculation with $T$ to enhance representation learning of the corresponding semantic primitives at the stage.

## 3.4 Training Strategy

The training objective comprises two components while employing a mixing coefficient $\lambda$:

$$\mathcal{L} = \mathcal{L}_{\text{CFM}} + \lambda\mathcal{L}_{\text{attn}}, \tag{10}$$

The first is conditional flow matching loss $\mathcal{L}_{\text{CFM}}$, which supervises denoising prediction, following the setting in MM-DiT. The other is $\mathcal{L}_{\text{attn}}$, which constrains the staged cross-attention by aligning injected semantic primitives with split text while regularizing attention behavior for stable semantic injection. It is composed of three components, weighted with empirically determined ratios of $\alpha = 0.6$, $\beta = 0.25$, and $\eta = 0.15$:

$$\mathcal{L}_{\text{attn}} = \alpha\mathcal{L}_{\text{inject}} + \beta\mathcal{L}_{\text{conv}} + \eta\mathcal{L}_{\text{mutex}}. \tag{11}$$

The first term, $\mathcal{L}_{\text{inject}}$, enforces the alignment between the cross-attention outputs and their target semantics at each activated injection stage:

$$\mathcal{L}_{\text{inject}} = \mathbb{E}_{t,x_0}\left[\sum_{i=1}^{3} \delta_i(t)\left(\text{CrossAttn}_i(c_{\text{base}}, c_{\text{inject}}^{(i)}) - c_{\text{target}}^{(i)}\right)^2\right]. \tag{12}$$

The second term, $\mathcal{L}_{\text{conv}}$, supervises the evolution of the attention signal-to-noise ratio (SNR) to ensure proper convergence timing:

$$\mathcal{L}_{\text{conv}} = \mathbb{E}_t\left[\sum_{i=1}^{3}\left(\text{SNR}_{\text{attn}}^{(i)}(t) - \text{SNR}_{\text{target}}^{(i)}(t)\right)^2\right]. \tag{13}$$

The third term, $\mathcal{L}_{\text{mutex}}$, penalizes spatial overlap between attention maps from different injection stages to maintain independence and prevent semantic interference:

$$\mathcal{L}_{\text{mutex}} = \mathbb{E}_t \left[ \sum_{i \neq j} \delta_i(t)\delta_j(t)\text{Overlap}(\text{AttnMap}_i, \text{AttnMap}_j) \right]. \tag{14}$$

Together, these components ensure stable multi-stage semantic injection by enforcing semantic precision, convergence regularity, and spatial exclusivity, which is crucial for achieving fine-grained semantic alignment in Split-Text Conditioning.

## 4  Experiment

### 4.1  Experiment Settings

**Impementation Details and Datasets**    To ensure fair comparison beyond the short-text bias of MM-DiT's original training data (CC12M [62] and ImageNet [63]), we adopt 100K samples from the SAM-LLaVA dataset [51], which contains semantically rich, long captions suitable for our split-text method. Extended training on this dataset yields MM-DiT 2B-E and MMDiT 8B-E. Following MM-DiT [26], we use 24 transformer layers with CLIP-L, CLIP-G, and T5-XXL as text encoders. Complete captions are parsed into hierarchical inputs via Qwen-Plus [61], with cross-attention incrementally injected at selected SNR-based denoising steps. Using the same training protocol, we obtain two variants of our models: DiT-ST M (2B) and DiT-ST L (8B). For evaluation, we follow CogView3 [64] and $\Delta$-DiT [65] to construct COCO-5K, assessing performance under varying text lengths and complexities, with results reported from our DiT-ST M and other competing method

**Evaluation Metrics**    We evaluate DiT-ST using both quantitative and qualitative metrics. For quantitative analysis, we adopt GenEval [66] to assess semantic comprehension, along with FID [67] and CLIPScore [68] to measure image quality and text-image alignment.

### 4.2  Main Results

In this section, we compare our method with MMDiT [26] and its variants as baselines, given architectural similarity and shared training setup. To further evaluate the generalizability of our split-text strategy in mitigating complete-text comprehension detect, we benchmark against competitive methods, including PixArt-$\alpha$ [51], Flux.1 Dev [50], and the state-of-the-art DreamEngine [52].

**Effectiveness of Split-text Caption Form**    Table 1 compares caption utilization forms in terms of CLIPScore and FID, showing that split-text caption leads to consistently better performance. By parsing prompts into organized semantic primitives, our method effectively mitigates the

Table 1: Comparison of the caption forms on COCO-5K.

| Caption Form | CLIPScore↑ | FID↓ |
|---|---|---|
| Original Caption | 31.68 | 24.61 |
| LLM-Enhanced Caption | 32.13 | 24.46 |
| Split-Text Caption | **34.09** | **22.11** |

complete-text defect, raises CLIPScore by 7.6 % and lowers FID by 10.2 % relative to the other two caption forms used on MM-DiT 2B-E, while still providing +6.1 % / –9.6 % gains over the LLM-Enhanced caption, underscoring that the observed gains are primarily attributable to the caption structure, rather than to external LLM intervention.

**Multi-Dimensional Semantic Comprehension Performance**    Table 2 reports comparisons on the GenEval benchmark, which evaluates six dimensions of semantic understanding: single-object recognition, multi-object reasoning, counting, color accuracy, spatial positioning, and attribute binding. Our method achieves a competitive overall accuracy of 69%, matching the state-of-the-art DreamEngine and closely approaching SDv3.5 Large (71%), despite a $4\times$ smaller parameters. Notably, our model outperforms the SDv3 Medium baseline across all subcategories, with marked advantages in multi-object reasoning(+0.14) and attribute binding(+0.10), indicating stronger semantic comprehension. Compared to Flux.1 Dev and DALL·E 3, which excel in specific areas, our approach offers more balanced and robust performance across all axes due to text-split strategy.

Table 2: Performance on GenEval↑ benchmark,while underlined is the second-best performance.

| Method | Parameter | Singel object | Two object | Counting | Colors | Position | Attribute Binding | Overall |
|--------|-----------|---------------|------------|----------|--------|----------|-------------------|---------|
| PixArt-$\alpha$ | 0.6B | 0.98 | 0.50 | 0.44 | 0.80 | 0.08 | 0.07 | 0.48 |
| DALL-E 3 | - | 0.96 | 0.87 | 0.47 | **0.83** | **0.43** | 0.45 | 0.67 |
| SDv3 Medium | 2B | 0.98 | 0.74 | 0.63 | 0.67 | 0.34 | 0.36 | 0.62 |
| Flux.1 Dev | 12B | 0.98 | 0.81 | **0.74** | 0.79 | 0.22 | 0.45 | 0.66 |
| SDv3.5 Large | 8B | 0.98 | 0.89 | 0.73 | **0.83** | 0.34 | 0.47 | **0.71** |
| Dream Engine | - | **1.00** | **0.94** | 0.64 | 0.81 | 0.27 | **0.49** | 0.69 |
| DiT-ST Medium | 2B | 0.99 | 0.88 | 0.70 | 0.75 | 0.36 | 0.46 | 0.69 |

Beyond GenEval, we employ CLIPScore and VQAScore [69] on the curated COCO-5K dataset. These two metrics provide a comprehensive evaluation: CLIPScore measures global semantic similarity, while VQAScore leverages a VQA model to directly probe for complex compositional alignment via a probabilistic "Yes/No" query. As results listed in Table 3, DiT-ST Medium attains a CLIPScore of 34.09, eclipsing competing methods by average 8.9 %, and even markedly surpassing SDv3.5 Large (32.74) by about 4.1%. On VQAScore, our DiT-ST Medium achieves 76.12, on par with the larger SDv3.5 Large, while surpassing SDv3 Medium (75.37) and other baselines. These results consistently validate that our split-text conditioning strategy enhances both compositional semantic fidelity and cross-modal alignment, enabling the model to generate images that more accurately capture the intended meaning of complex prompts while maintaining high perceptual quality.

Table 3: Performance of CLIPScore and VQAScore on COCO-5K.

| Method | PixArt-$\alpha$ | SDv3 Medium | Flux.1 Dev | SDv3.5 Large | DiT-ST Medium |
|--------|-----------------|-------------|------------|--------------|---------------|
| CLIPScore↑ | 32.58 | 31.31 | 31.52 | 32.74 | **34.09** |
| VQAScore↑ | 68.86 | 75.37 | 75.19 | **76.49** | 76.12 |

**Robust Performance to Caption Length**    Table 4 exposes marked length sensitivity in existing models: PixArt-$\alpha$ benefits from increasing caption length before plateauing, whereas SD-series models peak on mid-length prompts and deteriorate on longer ones (e.g., SD v3 Medium falls to 27.7 in the $[45, 55)$ subband). By contrast, our 2B-parameter model sustains a CLIPScore over 32 across all bins, rises to 35.81 for the longest captions, and secures the highest overall score (34.09), outperforming SDv3 Medium and SD v3.5 Large by 8.9% and 4.1%, respectively. These results highlight the robustness and effectiveness of our split-text conditioning strategy in handling variable-length and semantically complex prompts, enabling more stable text-image alignment across diverse input distributions while remaining parameters effcient.

Table 4: CLIPScore performance comparisons on various caption length in Selected COCO-5K.

| Method | Parameter | [10,15] | [15,25) | [25,35) | [35,45) | [45,55) | Average CLIPScore |
|--------|-----------|---------|---------|---------|---------|---------|-------------------|
| PixArt-$\alpha$ | 0.6B | 30.99 | 31.99 | 33.26 | 33.37 | 33.29 | 32.58 |
| SDv3 Medium | 2B | 31.91 | 32.69 | 33.37 | 30.86 | 27.73 | 31.31 |
| Flux.1 Dev | 12B | 30.59 | 31.16 | 32.36 | 31.08 | 32.83 | 31.52 |
| SDv3.5 Large | 8B | 32.17 | **32.96** | **34.10** | 32.93 | 31.53 | 32.74 |
| DiT-ST Medium | 2B | **32.28** | 32.47 | 33.84 | **34.76** | **35.81** | **34.09** |

## 4.3    Ablation Analysis

**Incremental Text Injection**    Incremental injection improves the text-split pipeline by scheduling semantic primitives at the appropriate timesteps. As Table 5 reports, this schedule rises CLIPScore by 3.9 % and simultaneously reduces FID by 7.3 %, decisively outperforming the single-shot alternative. The progressive delivery deepens semantic grounding, alleviates attention saturation and inter-token interference, enhancing both text alignment and visual fidelity.

Table 5: Comparison of incremental injection under CLIPScore and FID.

| Method | CLIPScore↑ | FID↓ |
|--------|------------|------|
| w/ injection | 34.09 | 22.11 |
| w/o injection | 32.81 | 23.85 |

**Hierarchical Caption Input**  Hierarchical caption input aligns naturally with the text-split strategy, as it organizes diverse semantic primitive types into a structured format for generation. To assess its effectiveness, we evaluate models under three input settings: original caption, LLM-Enhanced caption, and hierarchical input. As Table 6 shows, merely swapping the caption format to the hierarchical variant lifts CLIPScore by 3.6 % and reduces FID by 3 %

Table 6: Comparison among input strategies under CLIP-Score and FID Metrics on COCO-5K.

| Method | | CLIPScore↑ | FID↓ |
|---|---|---|---|
| SDv3 Medium | Original Caption | 31.31 | 24.66 |
| | LLM-Enhanced Caption | 31.95 | 24.46 |
| | Hierarchical Input | **32.42** | **24.05** |
| DiT-ST Medium | Original Caption | 31.68 | 24.61 |
| | LLM-Enhanced Caption | 32.13 | 24.43 |
| | Hierarchical Input | **32.81** | **23.85** |

for each architecture, with additional 2 % gains over the LLM-Enhanced captions. Crucially, SDv3 Medium enjoys the same absolute improvement as our own model, despite zero fine-tuning, demonstrating that the benefit is inherent to the representation and broadly transferable across models.

**More Ablation Analysis**  To further validate the effectiveness of our model and design choices, we conduct additional experiments on the orders of semantic primitive injection, timestep selection, hyperparameter sensitivity, and other ablation studies. Please refer to the appendix for more details.

## 4.4  Visualization

**High-Quality Generation Visualization**  Figure 6 showcases representative image samples generated by our DiT-ST, with rich visual details and strong semantic alignment across diverse prompts.

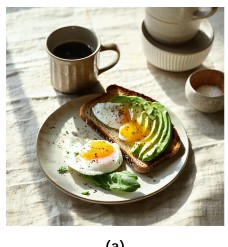 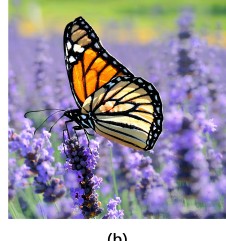 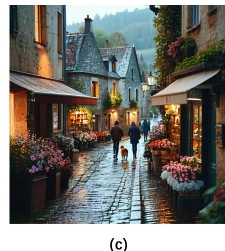 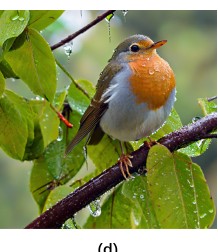

(a)            (b)            (c)            (d)

(a) A rustic breakfast withpoached eggs and avocado on sourdough toast, served on a ceramic platewith coffee, bathed in softmorning sunlight.
(b) A vibrant monarch butterfly with orange and black wings perched on a blooming lavender flower, surrounded by other purple blossoms.
(c) At dusk, villagers stroll leisurely along a charming cobblestone street adorned decorated with flower in the rain with their dogs.
(d) A small robin with a bright orange chest perches on arain-drenched branch, surrounded by wet green leaves.

Figure 6: High-quality generation visualization of DiT-ST Large.

**More Visualiztion**  We provide additional visual content, including results of different caption forms (e.g. Figure 1 (a)), performance against SDv3.5 (e.g. Figure 1 (b)) and other competing models as well as our high-quality images at multiple resolutions, which are available in the Appendix.

## 5  Conclusion

This paper introduces DiT-ST, a split-text conditioning framework that alleviates the complete-text comprehension defect in DiTs. It comprises three components. (*i*) Caption parsing, use LLMs to extract and organize primitives; (*ii*) Hierarchical input construction, construct split-text inputs and enrich input semantics; (*iii*) Incremental primitive injection, inject different primitives into appropriate denoising stages to improve stage-specific representation learning. Extensive experiments demonstrate the effectiveness of our split-text caption design and the excellent performance of our proposed DiT-ST. We hope this work will offer some inspiration to the diffusion community.

## Acknowledgement

This work is supported by the National Natural Science Foundation of China (No. 62576251, No. 62376198, and No. 62576247), the National Key Research and Development Program of China (No. 2022YFB3104700), and the Fundamental Research Funds for the Central Universities. The authors would like to thank Tianyu Wang and Jiu for generous support.

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

## A    Dataset Details

We further introduce the datasets used in our work from two aspects: training and evaluation.

For training, we observe that existing datasets used by baseline models MM-DiT [26] are suboptimal for complex caption-conditioned generation. Specifically, datasets like CC12M [62] have relatively short captions (average length 10.3), and ImageNet [63] provides only class-level supervision. Inspired by Pixart-$\alpha$ [51], which employs LLaVA to refine raw captions into high-information-density text, we adopt the SAM-LLaVA dataset provided by Pixart-$\alpha$ to enhance textual supervision. We select 100K pairs from SAM-LLaVA (available in our released CSV files) to further train MM-DiT 2B and 8B, aiming to improve their capacity on long-text generation and ensure fair comparisons.

For evaluation, we construct COCO-5K, following settings from CogView3 [64] and $\Delta$-DiT [65]. It comprises 5,000 image-text pairs sampled across multiple caption-length intervals to assess performance under varying textual complexity. Additionally, we report gFID scores based on COCO-30K to assess distribution-level image fidelity. Complete results are provided in Appendix D.

## B    Implementation Details

In terms of implementation, we focus on the settings of key thresholds and corresponding code-level realizations. We set the moving average window size $w$ to 3, allowing each $\widehat{\Delta}_t$ to consider the current and two preceding steps for better capturing the SNR trend. To ensure numerical stability when computing normalized differences, we set $\theta = 10^{-8}$. For identifying the convergence point, we use a convergence threshold $\tau = 10^{-4}$. During inference, we select 40 denoising steps from the full diffusion range (0–1000) via uniform random sampling. When determining the injection layers, we allow flexible injection intervals. For example, around timestep 50, we inject attribute primitives within a $\pm 10$ timestep range, while around timestep 400, we permit a broader window of $\pm 40$ timesteps, reflecting the varying stability of attention dynamics at different diffusion stages.

## C    Further Evaluation and Model Comparison

To reduce metric variance and ensure fair model comparison, we construct a fixed 30K image-text subset from COCO for consistent evaluation across architectures and training scales. We evaluate and compare our method against several representative baselines, including SDv3 Medium, SDv3.5 Large, Flux .1 Dev, Pixart-$\alpha$. The evaluation results, in terms of FID, are summarized in Table 7. On the fixed COCO-30K subset, our DiT-ST Large model **obtain the lowest FID scores**. At the 2B scale, DiT-ST Medium reaches 18.78, narrowly beating SD v3 Medium (18.82) by 0.2 %. With 8B parameters, DiT-ST Large achieves 17.16, improving on SD v3.5 Large (17.31) by 0.9 %, a modest yet consistent gain that shows split-text conditioning still reduces late-stage artefacts even on short COCO captions. Flux .1 Dev (32.10) and PixArt-$\alpha$ (27.35) post FIDs about 40% higher than DiT-ST Large. Flux, tuned for fast continuous-token sampling and lacking COCO fine-tuning, yields texture artefacts; PixArt-$\alpha$, trained on long, stylised captions, mismatches short, photorealistic domain. Thus, DiT-ST equals or surpasses SD baselines and clearly outperforms models trained on divergent data.

Table 7: Performance of FID on COCO-30K.

| Method | PixArt-$\alpha$ | SDv3 Medium | Flux.1 Dev | DiT-ST Medium | SDv3.5 Large | DiT-ST Large |
|--------|---------|-------------|------------|---------------|--------------|--------------|
| Param. | 0.6B | 2B | 12B | 2B | 8B | 8B |
| FID↓ | 27.35 | 18.82 | 32.10 | 18.78 | 17.31 | **17.16** |

## D    Ablation on Injection Order of Semantic Primitives.

In this experiment, we investigate how varying the injection order of semantic primitives, namely object, relation, and attribute, affects model performance. While our default configuration follows an object–relation–attribute order based on semantic granularity and generation stability, we test alternative permutations to examine the sensitivity of DiT-ST Medium to ordering strategies. Table 8 reveals the cost of premature information exposure. Advancing attribute tokens from their intended late-stage slot to attribute-relation-object or attribute-object-relation cuts CLIPScore to 29.41 and

31.10, drops of 14 % and 9 % from the default order. Early injection forces fine-grained colour and texture cues into an unstable attention landscape; later denoising steps overwrite or mis-bind these details, sharply reducing semantic fidelity.

Table 8: Comparison of different injection order for semantic primitives.

| Order | CLIPScore↑ |
|---|---|
| attribute-relation-object | 29.41 |
| attribute-object-relation | 31.10 |
| relation-object-attribute | 32.52 |
| relation-attribute-object | 30.16 |
| object-attribute-relation | 30.79 |
| object-relation-attribute (our setting) | **34.09** |

## E  Ablation on step selection strategy for semantic primitive injection.

We further investigate the impact of injection timestep choices for different semantic primitive types in the diffusion denoising process. Given that semantic primitives differ in abstraction level and generation dependency, we design a progressive injection schedule tailored to each type:

- Object tokens are injected at the early stages of denoising (timesteps 0, 10, and 20), where the model primarily focuses on establishing global layout and coarse structural elements. We experiment with these positions to compare their impact and identify the most effective timing for guiding the foundational composition of the generated image.

- Relation tokens are injected during mid-stage denoising. Specifically, we begin with timestep 25 as the midpoint of [0, 50], and extend to timestep 75 with five evenly spaced steps (30, 40, 50, 60, 70). We experiment with these positions to assess their impact and determine the optimal timing for enhancing spatial and compositional coherence.

- Attribute tokens are injected during the transition from semantic interpretation to visual detail refinement, when the model shifts focus from global structure to fine-grained features. To study their impact on visual modeling, we take the midpoint of the [50, 400] range as reference and select timesteps at 100-step intervals—specifically 200, 300, 400, 500, and 600—for evaluation.

This step allocation strategy reflects the assumption that lower-level semantics (e.g., object identity) should be introduced early to guide coarse synthesis, while higher-level or localized attributes benefit from later-stage injection when image fidelity and semantic detail are resolved.

As shown in Table 9, deviating from the default object-relation-attribute (O–R–A) schedule markedly impairs text–image alignment. Postponing object injection to $t = 10$–20 lowers CLIPScore by roughly 5–6 %, underscoring the need for early global cues. Advancing relation tokens to $t = 30$ reduces the score by about 4 %, as premature relational reasoning diverts attention from still-forming objects. The most severe loss, nearly 8 %, occurs when attribute tokens are introduced at $t = 200$; exposing fine-grained details before the scene is stabilised disrupts subsequent refinement as claimed in our main paper the question of premature information exposure. In aggregate, mis-timed injections can **degrade alignment by average 5%**, confirming that stage-aware scheduling is critical for maintaining semantic fidelity.

Table 9: Comparison of Different Step Selection Strategy by CLIPScore(CS).

| Step | O(10) | O(20) | R(30) | R(40) | R(60) | R(70) | A(200) | A(300) | A(500) | A(600) | Default |
|---|---|---|---|---|---|---|---|---|---|---|---|
| CS↑ | 32.71 | 32.24 | 32.57 | 32.73 | 32.76 | 32.45 | 31.26 | 31.75 | 32.96 | 32.83 | **34.09** |

## F  Ablation on Sliding Window Size

Table 10 shows that CLIPScore peaks at 34.09Our when the SNR-smoothing window is set to $w = 3$.A narrow window ($w = 1$) amplifies local noise, shifting the detected inflection point and trimming the score to 33.16 (-2.7 %). Expanding to $w = 2$ alleviates some volatility, yet still lags

the optimum at 33.55 (-1.6 %). Oversmoothing with $w = 4$ and $w = 5$ postpones relation-token injection; scores decline to 33.72 (-1.1 %) and 33.43 (-1.9 %), respectively. These results confirm that an intermediate window is essential—wide enough to suppress spurious curvature spikes yet narrow enough to capture the true SNR transition—thereby yielding the most consistent relation-level guidance and highest text–image alignment.

Table 10: Comparison of different sizes of sliding window.

| Window Size | CLIPScore↑ |
|---|---|
| 1 | 33.16 |
| 2 | 33.55 |
| 3 (our setting) | **34.09** |
| 4 | 33.72 |
| 5 | 33.43 |

## G   Ablation on Text Encoding Refinement.

As detailed in Section 3.2.2, our design includes a refinement for the classical text encoder of MM-DiT. Specifically, we fully utilize the previously wasted dimension capacity by incorporating the complete caption to enrich the overall semantic information of the input. To further validate the effectiveness of this refinement, we conduct an ablation experiment on our DiT-ST Medium model, trained for 5 epochs on 50K SAM-LLaVA dataset. The results are presented in Table 11. Without text encoding refinement, the model performance has declines of 0.62% in CLIPScore and 0.54% in FID metrics. The result indicate that although the improvement from adding complete text captions is modest, it indeed contributes positively to model performance without introducing additional computational or architectural overhead.

Table 11: Comparison of w/ and w/o Text Encoding Refinement.

| Method | CLIPScore↑ | FID↓ |
|---|---|---|
| w/ Text Encoding Refinement | **32.33** | **24.18** |
| w/o Text Encoding Refinement | 32.13 | 24.31 |

## H   Limitation

**The injection timestep selection operates at the batch level rather than the sample level.** This is because sample-level adaptation would significantly increase computational cost and reduce inference efficiency, coupled with the current lack of research on sample-level adaptation in the conditioning of denoising processes. Therefore, this study adopts a batch-level adaptation design as a compromise.

**The split-text caption may weaken cross-primitive semantic dependency.** Some special long-range dependencies (e.g., irony) could be split, leading to biased semantic understanding in the model. To mitigate this issue, DiT-ST introduces the text encoding refinement design, which incorporates original caption to compensate for the loss. Furthermore, existing works in the same field have not explored evaluating the semantic fidelity of special cases such as irony.

**The evaluation metrics lack human subjective assessment.** The evaluation of DiT-ST primarily relies on CLIPScore and FID, aligning with other works in the same filed. These metrics lack human perception, particularly in fine-grained details (e.g., color tone, detail consistency). To compensate for this limitation, similar to other prior works, we provide extensive visualizations to facilitate subjective evaluation by readers.

# I  More Visual Comparison Across Caption Forms

|  Original Caption
(Complete-Text) | LLM-Enhanced Caption
(Complete-Text) | Our Caption
(Split-Text) |
|---|---|---|

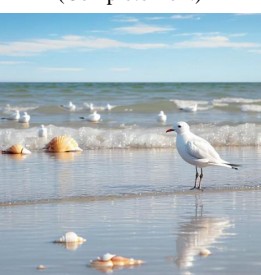 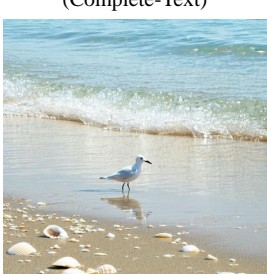 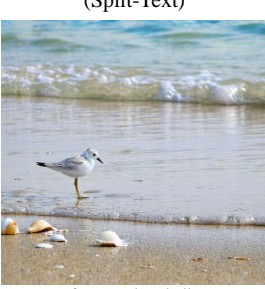

**Original:** A quiet beach with soft sand, scattered white seashells. One small bird stands alone near the water, its feet in the shallow waves, looking down at its reflection. Gentle waves roll in behind it.
**LLM-Enhanced:** There is a quiet beach with soft sand and scattered white seashells. A small bird stands alone near the water, looking at its reflection.
**Our:** [OBJECT] beach. sand. seashells. bird. waves. [RELATION] beach has sand and seashells. bird stands near water. feet are in waves. bird looks at its reflection. waves roll in behind the bird. [ATTRIBUTE] The beach is quiet. The sand is soft. The seashells are white and scattered. The bird is small and alone. The waves are gentle and shallow.

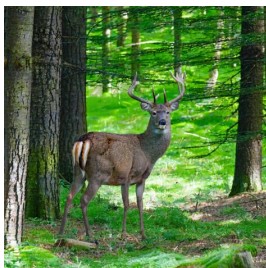 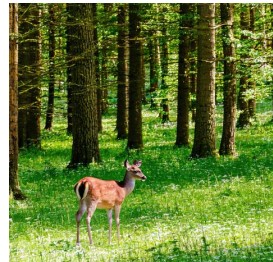 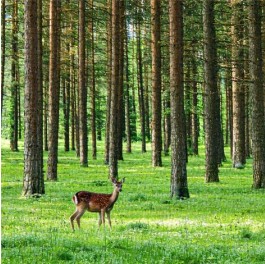

**Original:** In a quiet forest filled with tall pine trees, a young deer with dots on its back stands still in the soft moss underfoot.
**LLM-Enhanced:** There are a quiet forest, tall pine trees and a young deer. The deer stands still on soft moss floor.
**Our:** [OBJECT] forest. trees. deer. moss. [RELATION] forest has tall pine trees. deer stands in moss. [ATTRIBUTE] The forest is quiet. The trees are tall and pine. The deer is young and with dots. The moss is soft.

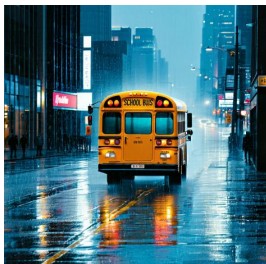 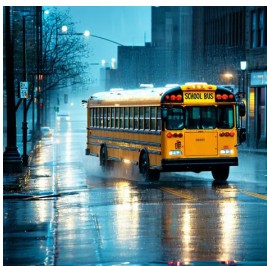 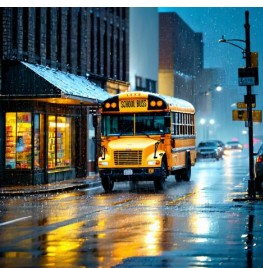

**Original:** A yellow school bus drove through a rainy city street at night. The damp road reflects the yellow shop lights on roadside, heavy rain is falling.
**LLM-Enhanced:** There is a yellow school bus driving through a rainy city street at night. The wet road reflects the yellow lights of nearby shop lights. The rain falls heavily and steadily.
**Our:** [OBJECT] school bus. street. shop. lights. rain. [RELATION] bus drives on damp street. shop has lights. street reflects lights. [ATTRIBUTE] The bus is yellow. The street is damp. The lights are yellow. The rain is heavy.

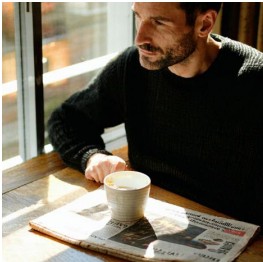 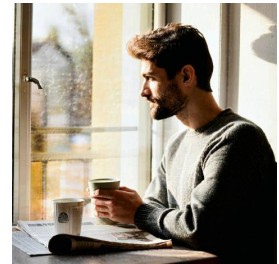 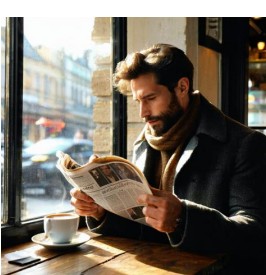

**Original:** At a morning café, a man was flipping through newspaper by the window, with warm sunlight shining across the wooden table and a cup of coffee beside him.
**LLM-Enhanced:** There is a man sitting by the window in a morning café, reading newspaper. Warm sunlight shines across the wooden table, and a cup of coffee beside him.
**Our:** [OBJECT] café. man. newspaper. window. sunlight. table. coffee. [RELATION] man sits by the window. man flips through newspaper. sunlight shines on the table. coffee is beside the man. [ATTRIBUTE] The café is morning. The table is wooden. The sunlight is warm.

| Original Caption (Complete-Text) | LLM-Enhanced Caption (Complete-Text) | Our Caption (Split-Text) |
|---|---|---|

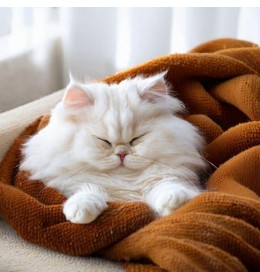 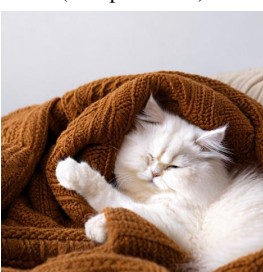 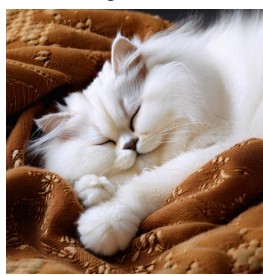

**Original:** A furry white cat quietly fell asleep, nestled in a warm, brown blanket adorned with delicate embroidered floral patterns and raised stitching.
**LLM-Enhanced:** There is a furry white cat sleeping in a warm brown blanket. The blanket is decorated with delicate embroidered floral patterns and raised stitching.
**Our:** [OBJECT] cat. blanket. patterns. stitching. [RELATION] cat sleeps in blanket. blanket is adorned with floral patterns. blanket has raised stitching. [ATTRIBUTE] The cat is furry and white. The blanket is warm and brown. The floral patterns are delicate and embroidered. The stitching is raised.

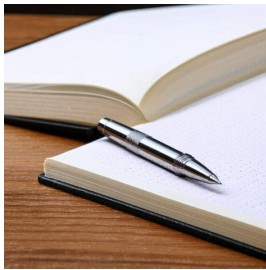 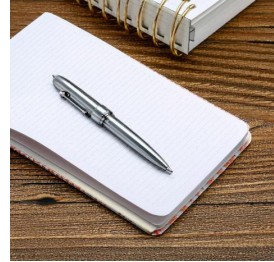 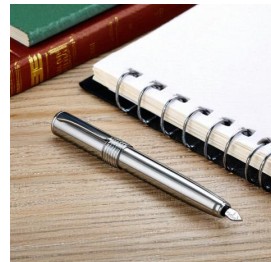

**Original:** A silver pen without its cap rests on a wooden table beside a blank white spiral-bound notebook. A few other notebooks also lie nearby.
**LLM-Enhanced:** There is a silver pen without cap resting on a wooden table next to a blank white spiral-bound notebook. Several other notebooks are scattered nearby.
**Our:** [OBJECT] pen. table. notebook. [RELATION] pen rests on table. pen is beside notebook. other notebooks lie nearby. [ATTRIBUTE] The pen is silver and uncapped. The table is wooden. The notebook is blank, white, and spiral-bound.

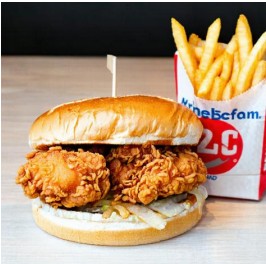 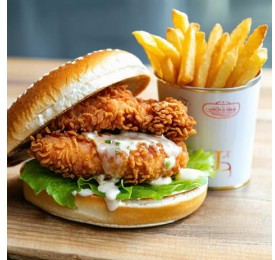 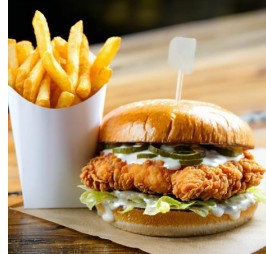

**Original:** A crispy fried chicken burger with lettuce, pickles, and cream sauce on top, placed beside a container of golden French fries on a wooden table.
**LLM-Enhanced:** There is a crispy fried chicken burger topped with lettuce, pickles, and cream sauce, sitting next to a container of golden French fries on a wooden table.
**Our:** [OBJECT] burger. lettuce. pickles. cream sauce. fries. container. table. [RELATION] burger has lettuce, pickles, and cream sauce. burger is beside fries. fries are in a container. burger and fries are on the table. [ATTRIBUTE] The burger is crispy and fried. The lettuce, pickles and cream sauce is on top. The fries are golden. The table is wooden.

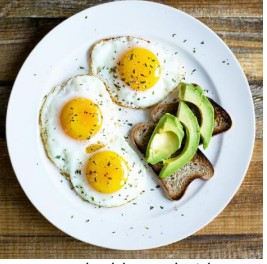 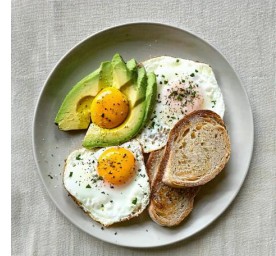 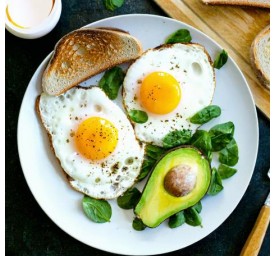

**Original:** A healthy meal with two egg yolk facing fried eggs, fresh half cut avocados, and a few slices of toasted whole wheat bread, lightly seasoned with herb.
**LLM-Enhanced:** There is a healthy meal consisting of two sunny-side-up eggs, fresh halved avocados, and a few slices of toasted whole wheat bread, lightly seasoned with herb.
**Our:** [OBJECT] meal. eggs. avocados. bread. herb. [RELATION] meal includes eggs, avocados, and bread. meal is seasoned with herb. [ATTRIBUTE] The meal is healthy. The eggs are fried and yolks up. The avocados are fresh and halved. The bread is whole wheat and toasted.

Figure 7: More comparisons of different caption form and corresponding performances.

# J  More Visual Comparison Across Different Models

In this section, we present more qualitative comparisons across different models. Specifically, we employ our large-scale model DiT-ST Large and compare it with the baseline MM-DiT 8B-E under identical prompt conditions. As shown in Figure 8, DiT-ST Large produces more compositionally coherent and semantically aligned results, especially in complex scenes with diverse attributes.

SDv3.5 Large          Our DiT-ST Large                    SDv3.5 Large          Our DiT-ST Large

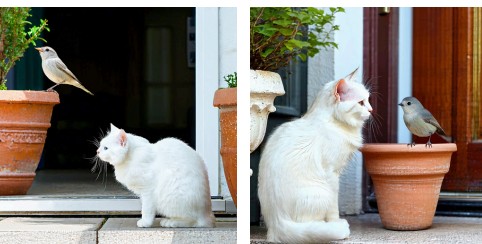

A cat sits calmly on the table in front of a keyboard and a monitor, scattered papers, and a few books. Through the window is green grass in a touch.

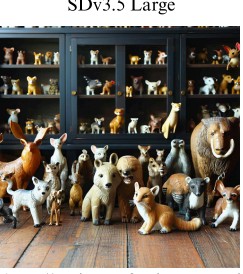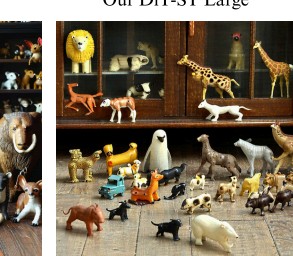

A collection of vintage animal figurines arranged on a wooden floor in front of a dark wooden cabinet filled with more toy animals.

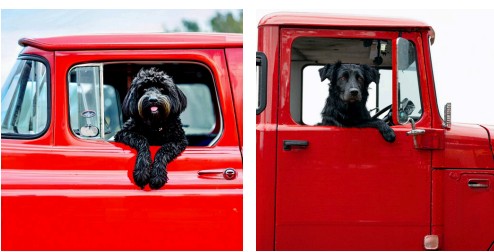

A white cat gently stared at a bird perched on a clay pot before the door.

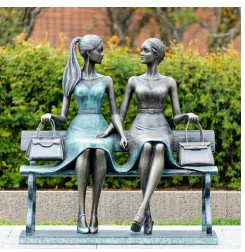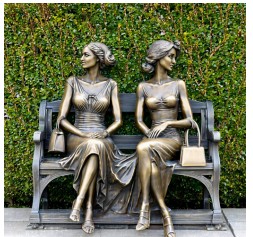

A pair of elegant bronze women states sits on a bench, each holding a handbag in front of a lush green hedge.

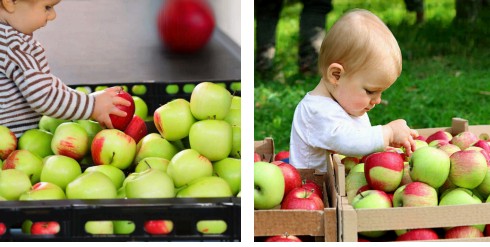

A black dog sitting in the driver's seat of a bright red vintage truck, looking out of the open window with its paw resting on the door.

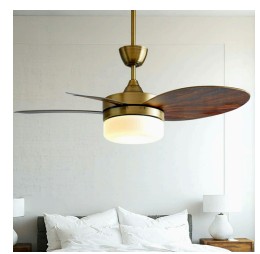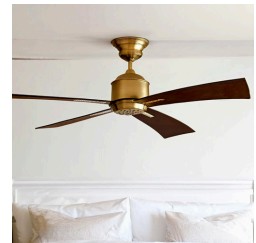

A brass ceiling fan with wooden blades and a built-in light fixture, mounted above a white bed in a minimal, modern bedroom.

A little baby is standing near to the apple boxes and touching the apples, the apples are mixed of green  and red in color.

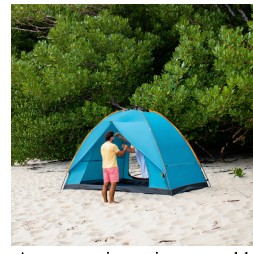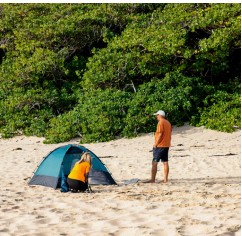

A woman is setting up a blue tent on a sandy beach near dense green foliage with a man in yellow T-shirt watching by.

Figure 8: Comparisons between MM-DiT 8B-E (left) and our DiT-ST Large (right)

We further compare DiT-ST Large with several state-of-the-art text-to-image generation models, including PixArt-$\alpha$, Flux, and HunyuanDiT. As illustrated in Figure 10, our model demonstrates competitive or superior visual quality, with stronger object fidelity, spatial consistency, and semantic expressiveness.

| Our DiT-ST Large | Flux .1 Dev | PixArt-α | HunyuanDiT |
| --- | --- | --- | --- |

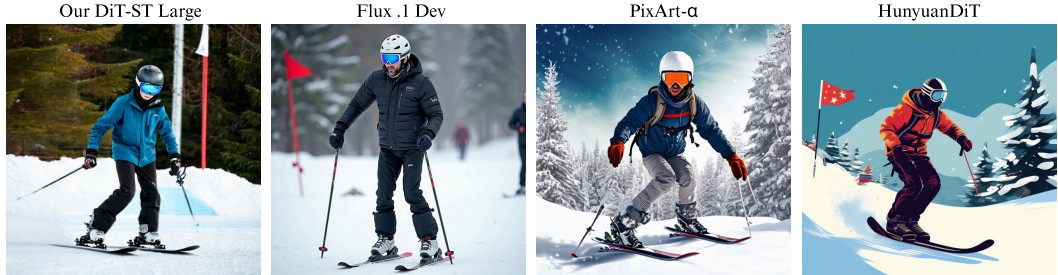

A person is skiing on snow, wearing gloves, a helmet, goggles, and a jacket, with a flag and trees in the background.

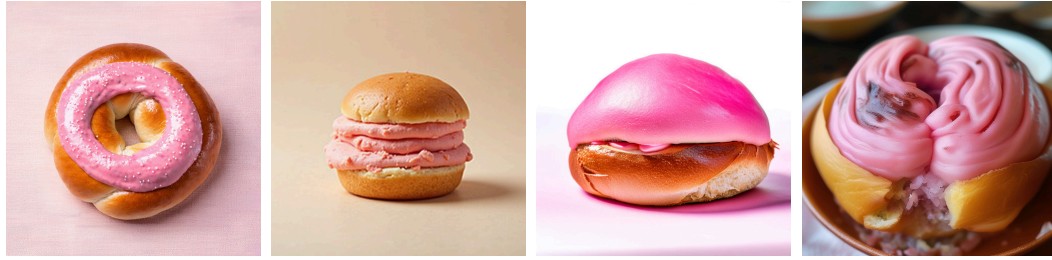

A bread roll with a golden, crispy exterior, coated in pink icing and sprinkled with white sugar pearls.

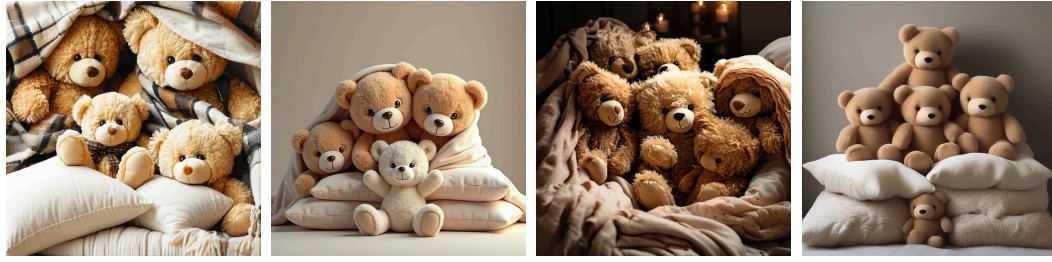

A group of teddy bears is wrapped in a blanket, with pillows underneath.

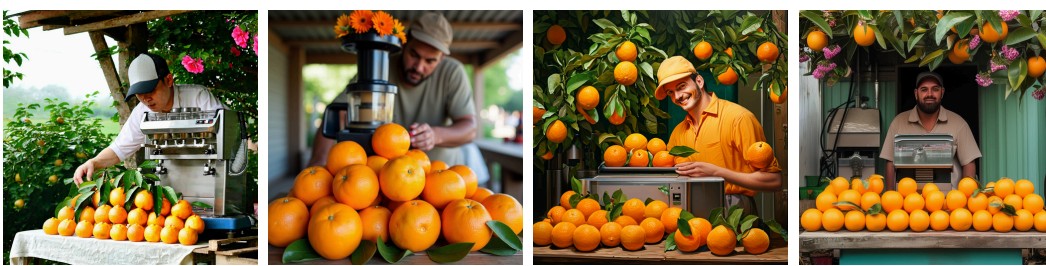

Under a shed with flowers, a man wearing a cap is working behind a table with piles of oranges and juice machine.

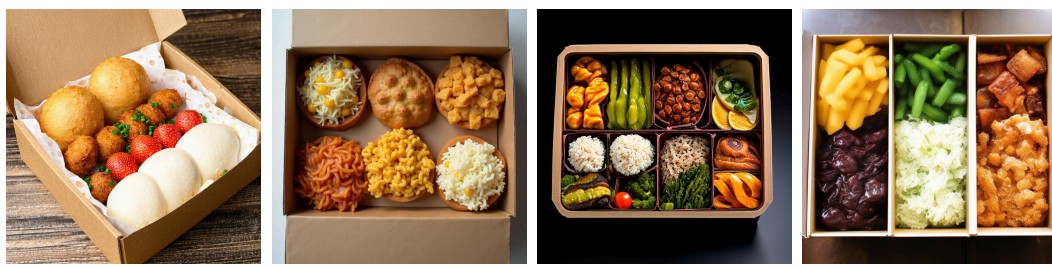

The box contains three types of food, with many fried items placed on a grease-absorbing paper lining.

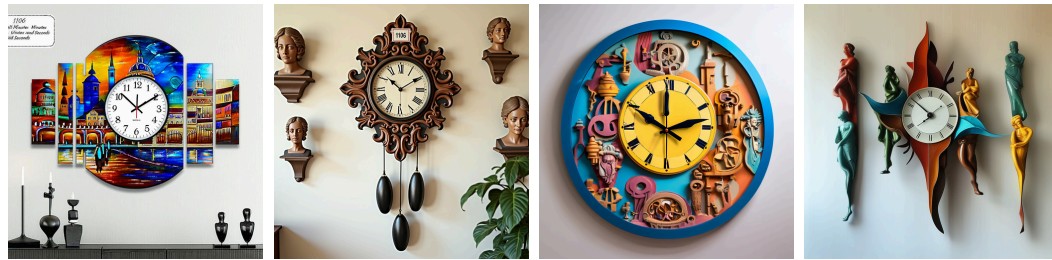

A multi-colored wall clock is mounted on the wall, surrounded by a few sculptures, a label above reads "1106."

Figure 9: Comparisons among DiT-ST Large, Flux, PixArt-α and HunyuanDiT.

# K    More High Quality Images Generated by Our DiT-ST Large

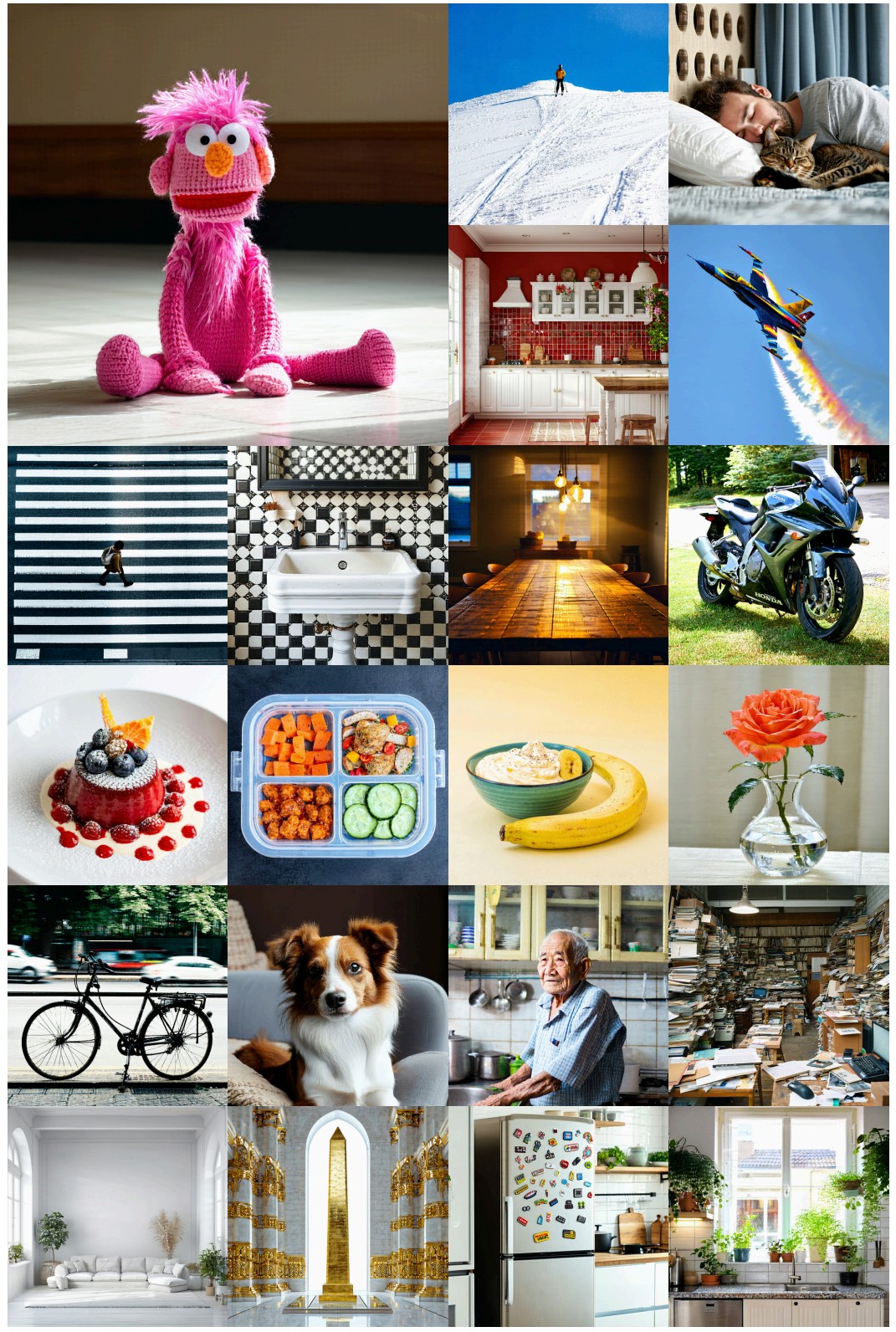

Figure 10: High-quality 1024×1024 images generated by our DiT-ST Large

Besides the standard 1024×1024 images shown in Figure 10, we further generate multi-scale samples put in Figure 11, confirming that our DiT-ST Large maintains visual fidelity and semantic alignment across different resolutions and formats.

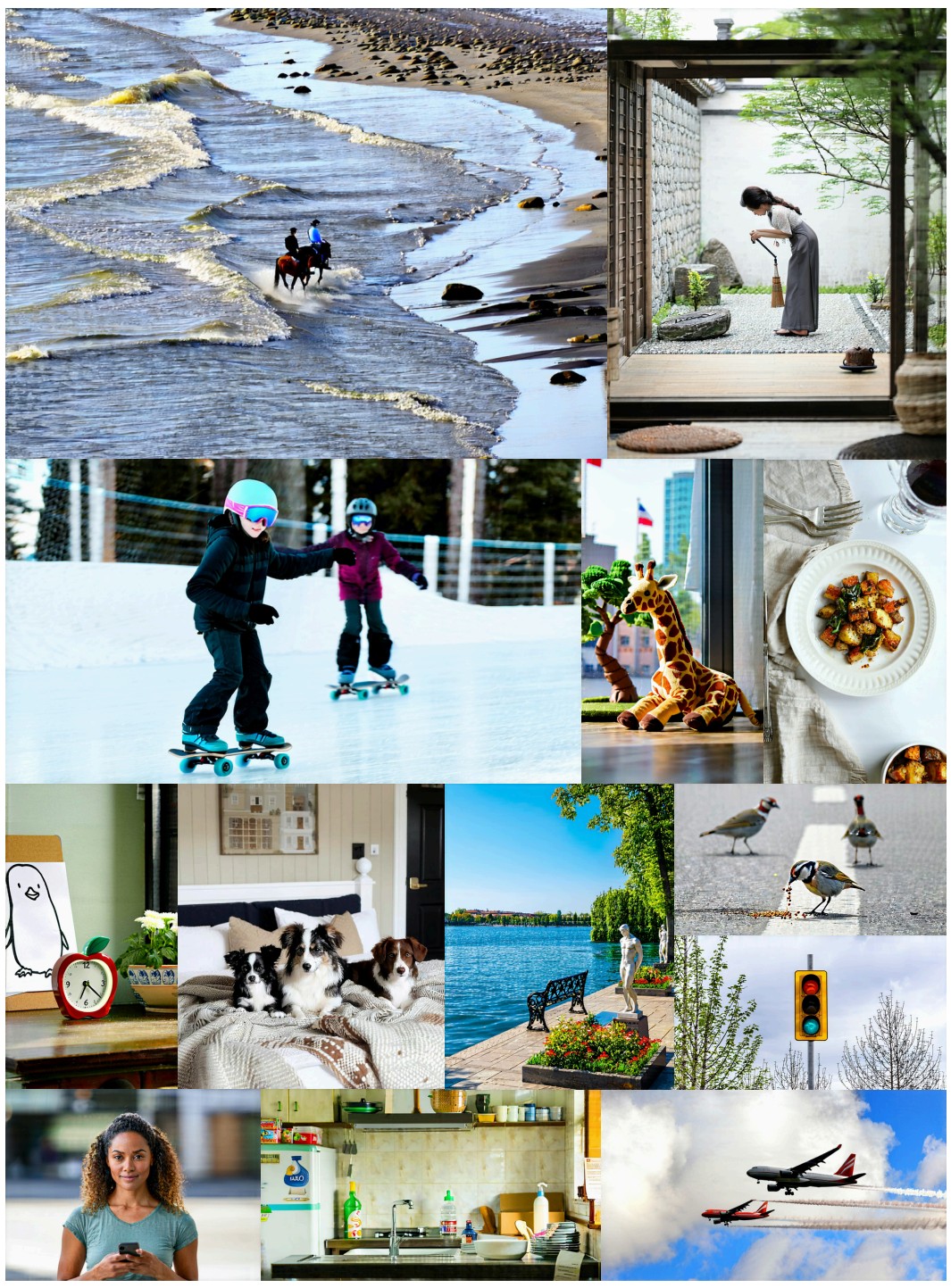

Figure 11: High-quality and multi-scale generation results by our DiT-ST Large

# NeurIPS Paper Checklist

