# OpenReview forum: "Enhancing Text-to-Image Diffusion Transformer via Split-Text Conditioning"
_NeurIPS.cc/2025/Conference — NeurIPS 2025 poster_

### Official Review · Reviewer_4VWJ · 2025-06-30

**Clarity:** 3
**Significance:** 2
**Originality:** 3
**Rating:** 4
**Confidence:** 4

**Summary:**

This paper proposes a text-to-image generation method using split-text as a condition. Existing text-to-image models struggle to comprehend complex semantic information, so the authors attempt to refine the text using large language models, constructing multiple complex sub-texts based on three dimensions: object, relation, and attribute. The authors also note that these three types of prompts have different impacts on the model during various stages of denoising in the diffusion model, leading to the design of a staged injection mechanism. Experiments show that this design effectively improves the model's ability to follow prompts.

**Questions:**

Please refer to the weakness.

**Ethical Concerns:**

["NO or VERY MINOR ethics concerns only"]

**Final Justification:**

After reading the author's rebuttal, my concerns have been addressed. I choose to raise my score.

**Limitations:**

yes

**Quality:**

2

**Strengths And Weaknesses:**

Strengths:

1. The author's use of a graph to model the text is an interesting approach. This method of normalizing the prompt can be helpful for weak text encoders.

2. The author conducted experiments on multiple models to further validate the effectiveness of the approach.

3. The paper is easy to understand.

Weaknesses:

1. Due to the limited text understanding capabilities of models like CLIP and T5, many existing works use powerful LLMs as text encoders for diffusion models[1,2,3]. Can this design further improve performance when used with LLMs with stronger understanding abilities?

2. In the paper, the authors inject object, relation, and attribute into the denoising process in three separate stages. However, since object and relation jointly determine the layout of the generated image, could using only the object prompt in the early denoising stages affect the accuracy of the relation, potentially leading to extra objects or incorrect relationships? The authors should analyze the output results from denoising with different prompts.

3. Regarding the cost analysis of Injection Timestep Selection, for different diffusion models, will these injection timesteps differ significantly, and what kind of time and resource overhead will this bring?

4. The evaluation benchmarks are weak. ClipScore and FID do not reflect the model's understanding of complex prompts. The authors should use additional benchmarks with LLMs as evaluators for ablation studies. They should also provide more showcases to demonstrate that the fine-tuned model has improved prompt comprehension.

5. Using a graph to model prompts lacks completeness. Some prompts don't contain fully defined semantic information, which is common in platforms like MidJourney. For a prompt that doesn't specify an object, what kind of output will the model generate?

[1] https://arxiv.org/abs/2406.11831

[2] https://arxiv.org/abs/2410.10629

[3] https://arxiv.org/abs/2505.10046

---

> ### Author Rebuttal · Authors · 2025-07-29
>
> Hi, Reviewer 4VWJ. We sincerely appreciate your careful review and so many professional and constructive comments, especially providing many useful and insightful references. We value each of them and provide responses below:
>
> >**W1: Powerful LLMs as encoders**
>
> Thank you for this interesting question. We reference these three works, with particular focus on the Complex Human Instruction from [2] to enhance image-text alignment. We conduct experiments using Qwen2-VL-1.5B and Gemma2 as text encoders. (50K dataset, 5 epochs & 20 epochs, base model: DiT-ST Medium)
>
> |TextEncoder|CLIPScore↑|FID↓|
> |-|-|-|
> |5 epochs|||
> |Ours|32.33|24.18|
> |Replaced by Qwen2-VL-1.5B|33.01|23.78|
> |Replaced by Gemma2|32.93|23.81|
> |20 epochs|||
> |Ours|33.51|23.37|
> |Replaced by Qwen2-VL-1.5B|34.74|21.64|
> |Replaced by Gemma2|34.59|21.75|
>
> As shown in the table, replacing our original text encoder with LLMs for 5 epochs training yields slight improvements in both CLIPScore and FID. Considering the short training duration, the model performance may be unstable and cannot provide strong evidence. Therefore, we further tested the model performance with 20 epochs of training. We found that as the scale increases, the effects become more obvious, indicating that using LLMs as text encoders can further improve generation quality. This is a very interesting and worthwhile question for deeper investigation. Thank you for providing this valuable insight.
>
> &nbsp;
>
> >**W2: Analysis of different injection processes**
>
> Thank you for your inspiration. Although we discussed the impact of different injection orders on performance in Table 8 in the appendix, we acknowledge that as you mentioned, object and relation jointly determine the layout of the generated image. Therefore, we supplement the following experiments based on Table 8: We replace the object injection at t=0 with 0.5object+0.5relation, 0.8object+0.2relation, and 0.2object+0.8relation respectively, to explore the joint effect of object and relation at the beginning of denoising.
>
> |Injection Order|CLIPScore↑|FID↓|
> |-|-|-|
> |O-R-A (ours)|34.09|22.11|
> |0.5O+0.5R|33.37|23.52|
> |0.8O+0.2R|33.85|23.84|
> |0.2O+0.8R|32.23|24.28|
>
> Surprisingly, the experimental results are counterintuitive. The effect of jointly injecting object and relation at the beginning of denoising is not as good as injecting object alone. Due to time constraints, we have not yet found the clear reason for this phenomenon. We speculate that this should be closely related to the semantic entanglement we discussed in the introduction, and ensuring independent modeling of objects at the beginning is crucial. Furthermore, for the 1000-step denoising process of SD, injecting object at t=0 and relation at t=50 can both be viewed as early-stage injection. For our O-R-A injection order, it can both ensure that object and relation jointly determine the layout of the generated image in the early stage, while also taking into account the decoupled modeling of object concepts, thus achieving better results. This is a very interesting topic, and we will continue to explore it in the future to verify the above speculation. Thank you again for your inspiration.
>
> &nbsp;
>
> >**W3: Injection timestep selection and overhead analysis**
>
> +**injection timestep selection**
>
> To validate the generality of our cross-attention and SNR-guided injection strategy, we conduct experiments on two additional architectures: Flux .1 Dev and HunyuanDiT-v1.2. By analyzing cross-attention convergence and SNR curves, we determine the O-R-A injection steps for each model and compare them with alternative steps. As shown below, injecting at the identified steps consistently yields the best performance, while other steps lead to degradation, confirming the method’s robustness.
>
> |Flux Step|O(10)|O(20)|R(20)|R(30)|R(50)|R(60)|A(100)|A(200)|A(400)|A(500)|default|
> |-|-|-|-|-|-|-|-|-|-|-|-|
> |CLIPScore |32.56|32.38|32.40|32.46|32.49|32.31|31.51|31.89|32.64|32.56|33.25|
>
> |Hunyuan Step|O(10)|O(20)|R(80)|R(90)|R(110)|R(120)|A(300)|A(400)|A(600)|A(700)|default|
> |-|-|-|-|-|-|-|-|-|-|-|-|
> |CLIPScore |32.37|32.22|32.47|32.55|32.50|32.31|31.76|32.08|32.48|32.41|32.93|
>
> Although the specific injection positions vary with model architectures—e.g., Flux (0, 40, 300) shifts earlier due to faster convergence, while Hunyuan (0, 100, 500) shifts later—our strategy remains effective across all tested models.
>
> +**overhead analysis**
>
> Before training, we determine Attribute and Relation injection steps via a one-time offline analysis, enabling phase-wise semantic conditioning without extra supervision or dynamic control.
>
> Specifically, for the Attribute injection timestep \$s\_{attr}\$, we detect the earliest point where cross-attention stabilizes. For each sample and every layer \$m=1,\dots,M\$ and head \$h=1,\dots,H\$, we compute the normalized difference of the cross-attention matrix \$Attn\_t^{(m,h)} \in R^{Q \times K}\$ at step \$t\$ (see Eq. 5), average across heads and layers, apply a moving average over the batch, and use thresholding to determine \$s\_{attr}\$. The computational complexity is \$T\_{s\_{\mathrm{attr}}} = O(N \times S \times M \times H \times QK)\$, where \$N\$ is the sample size, \$S\$ the diffusion steps, \$M\$ and \$H\$ the numbers of layers and heads, and \$Q\$, \$K\$ the attention dimensions. As this one-time offline process uses few samples (e.g., \$N=100\$), its cost is negligible relative to a training epoch.
>
> For Relation, we leverage the static SNR (Signal-to-Noise Ratio) curve provided by the diffusion scheduler, defined as  $\mathrm{SNR}(t) = \alpha_t^2 / \sigma_t^2$,  and compute its discrete curvature over the interval $[s_{\mathrm{attr}}, S]$ to locate the turning point corresponding to the optimal Relation injection timestep $s_{\mathrm{rel}}$ (see Eqs. 8 and 9). This method requires no model inference or input data, and only involves simple discrete operations over a one-dimensional sequence of length  $S' = S - s_{\mathrm{attr}}$.  Its computational complexity is: $T_{s_{\mathrm{rel}}} = O(S - s_{\mathrm{attr}})$. Since it only performs basic operations over a short vector of scalar values, the runtime is significantly lower than that of the matrix-based analysis used for $s_{\mathrm{attr}}$, and can be considered negligible.
>
> &nbsp;
>
> >**W4: More comprehensive evaluation**
>
> We fully agree with you. CLIPScore and FID have inherent limitations, as they cannot adequately evaluate image naturalness, fidelity, and rationality. Therefore, they cannot reflect the model's understanding of complex captions. We additionally adopted GenEval for more comprehensive evaluation, such as semantic consistency, fidelity, clarity, diversity, and plausibility. While many other metrics exist, these three are the most widely adopted in the field. Unfortunately, when seeking to conduct broad comparative analysis, the number of applicable universal metrics becomes limited. Inspired by Reviewer `6Z8X`, VQAScore can serve as a general benchmark similar to GenEval. The following presents the evaluation of our method and other baselines on VQAScore.
>
> |Method|VQAScore(\*)↑|VQAScore↑|
> |-|-|-|
> |SDv3 Medium|75.37|58.56|
> |SDv3.5 Large|76.49|60.77|
> |PixArt-α|68.86|50.83|
> |Flux.1 Dev|75.19|55.15|
> |DiT-ST Medium|76.12|62.73|
> |DiT-ST Large|76.25|64.30|
>
> \* denotes the simplified VQAScore used in [r1]. The other VQAScore is standard. We utilize the semantic primitives from the split-text to construct our standard VQA questions: *"Is [OBJECT] visible in the image?" "Does the [OBJECT] have [ATTR]?" "Does the [OBJECT] appear [REL] the [OBJECT]?"*
>
> We can see that in the simplified VQA, the gap between models is small. However, in the standard VQA, DiT-ST shows a clear lead, demonstrating better capability in handling complex semantics and captions. Additionally, we provide extensive visual showcases in **Figures 7-11 in the appendix** for your evaluation. Once again, thank you for your suggestion and inspiration!
>
> [r1] Progressive Prompt Detailing for Improved Alignment in Text-to-Image Generative Models, arxiv 2025
>
> &nbsp;
>
> >**W5: Output of semantically incomplete captions**
>
> For prompts that lack explicit objects, we view this as a data-related issue rather than a limitation of our method. Our model is designed to parse complex prompts into structured semantic primitives and generate faithful images when sufficient information is provided. When a prompt omits core content (e.g., just like“dreamlike atmosphere” or “golden hour”), our method naturally falls back to global style or scene-level cues without object-level construction. Importantly, this challenge is not unique to our approach—existing models targetting at solving complex prompts would also behave similarly under such under-specified prompts. Addressing such cases typically requires prompt enrichment or additional context, rather than changes to the underlying generation framework.
>
> &nbsp;
>
> &nbsp;
>
> Once again, we sincerely thank you for dedicating your valuable time to review our submission and for providing such insightful and professional suggestions. We are fully committed to incorporating the analyses and experiments you recommended into the final version, as a way to respect your time and effort. To be frank, you are a pivotal reviewer whose feedback may play a decisive role in the outcome of this submission, and we genuinely hope to earn your encouragement and support. Due to rebuttal policy restrictions, we regret that we are currently unable to provide the relevant visualizations for your review. However, if you have any further questions or feel that any of our explanations are unclear, please don’t hesitate to let us know—we would be more than happy to provide additional detailed clarifications. Thank you once again!
>
> &nbsp;
>
> Best wishes and regards,
>
> All authors of Submission 510

---

> > ### Comment · Reviewer_4VWJ · 2025-08-05
> >
> > After reading the author's rebuttal, my concerns have been addressed. I will raise my score to ba.

---

> ### Author Response · Authors · 2025-08-05
> **Appreciation for Your Review and Feedback**
>
> We sincerely thank you for carefully reading our rebuttal and for your encouraging decision to adjust the rating.We are delighted that our additional analyses and clarifications have addressed your concerns.
>
> Additionally, we are especially grateful for your insightful suggestion regarding leveraging LLMs to further enhance the capability of text encoders. This has provided us with significant inspiration, and we plan to continue exploring this direction in our future work.
>
> Once again, we deeply appreciate the time and effort you have dedicated to reviewing our submission. Your professional feedback has been invaluable in helping us improve this work, and we genuinely hope that this submission will ultimately achieve a positive outcome which lives up to your effort and trust.
>
> Best wishes and regards,
>
> All authors of Submission 510

---

### Official Review · Reviewer_kbep · 2025-07-01

**Clarity:** 3
**Significance:** 2
**Originality:** 2
**Rating:** 5
**Confidence:** 4

**Summary:**

The paper approaches to improve T2I diffusion models by splitting text captions into smaller, independent, sentences that contain different (hierarchical) semantic primitives, e.g., general image layout vs vs details for specific objects. These primitives are then injected into the diffusion model at different timesteps, starting with more general information at the beginning of the sampling process and ending with detailed object information at the end of the sampling process. The evaluation shows improved scores on GenEval, CLIP score, and FID.

**Questions:**

The approach needs additional processing both at training and test time. It would be interesting to do some ablations comparing the effect of different ways of doing the split-text caption construction. It seems relatively involved right now and sounds somewhat brittle.

Similarly, there is some extra design involved in how and when to inject the different kinds of captions. One interesting baseline would be to just feed the split-text captions into a "normal" pretrained models (without any finetuning) at the given timesteps to check the kinds of improvements this would lead to (if any).

It also seems like right now the conditioning also always includes (a projection of) the full caption. How important is that? Does it still work if it only conditions on the split-text conditions?

How will this approach affect potential down-stream tasks (such as editing) which would be finetuned from a pretrained T2I model?

**Ethical Concerns:**

["NO or VERY MINOR ethics concerns only"]

**Final Justification:**

Thank you for the detailed rebuttal. It has addressed my concerns and I have updated my rating to accept.

**Limitations:**

The finetuning of the models was done with a dataset that only contains 100K examples. This is relatively small-scale and injecting human biases is likely to help here. It is not clear to me that we would see the same kinds of improvements when a general diffusion model is trained on a large-scale dataset of longer and more complex prompts. Table 2 also shows that models trained on a dataset of more variety also perform well in most settings. It is not clear that using the split-text condition is really helpful compared to training on a larger dataset with diverse, long, and detailed captions.

Additionally, the approach requires some specific design choices (e.g., how exactly to create the split-text captions, when and how to inject them, etc) and it is not obvious to me that this will easily generalize across different datasets and model architectures.

**Quality:**

2

**Strengths And Weaknesses:**

The paper addresses an important topic of improving text-image alignment in diffusion models. The breaking down of a complex instruction into various smaller sub-instructions makes intuitive sense. Structuring the creation of the sub-instructions with the well-known behavior of diffusion models to first generate broad layouts and later on focus on semantic details also makes sense.

The paper finds ways to automatically address these, by automatically creating sub-captions and proposing an automated way to find when to inject them into the model. That being said, the evaluation is only done on one kind of model so it is not clear if this transfers easily to other models and whether the injection approach also works on other models.

However, there are many ways to split a caption into sub-sections and it's not clear that the one proposed here is the best one or transfers to different models. It might also depend on the kind of data that a model is trained on.

This also leads to extra compute requirements, both at train and at test time, since the input captions have to be processed to the split-text format. It also requires additional finetuning of the model, though the approach could also work without finetuning (though this is not tested in the paper I think).

---

> ### Author Rebuttal · Authors · 2025-07-29
>
> Hi, Reviewer kbep. We sincerely appreciate your careful review and so many constructive comments, especially for pointing out some limitations that we had not previously recognized. We value each of them and provide responses below:
>
> >**W1: Transferability to other models**
>
> We appreciate you highlighting this issue. We recognize that this experiment is essential for validating the transferability and generality of our method. Therefore, we conduct a comparative study. Our method initially employs SD as the base model, now, we expand our evaluation to include other Diffusion Transformer models. We apply our method to Flux .1 Dev and HunyuanDiT to assess its effectiveness across other models.
>
> |Method|CLIPScore↑|FID↓|
> |-|-|-|
> |SDv3 |30.84|24.77|
> |Ours-SDv3 (i.e. DiT-ST Medium)|32.33|24.18|
> |Flux|31.52|26.69|
> |Ours-Flux|33.25|25.16|
> |HunyuanDiT|32.04|25.24|
> |Ours-HunyuanDiT|32.93|25.07|
>
> Experiment setting: 50K dataset, 5 epoch, base model: SDv3 Medium. The experiment results demonstrate that applying our method to other models, namely Flux and HunyuanDiT, also yields good performance. This demonstrates the effective transferability of our method across same model architectures Given that these models share a common transformer-based architecture, they can similarly leverage cross-attention mechanisms for injection.
>
> &nbsp;
>
> >**W2 & Q1: Different caption splitting strategies**
>
> Thank you for raising this question. First, our caption splitting strategy can be transferred to other models and achieves good performance (as jointly demonstrated by Table 5, your W1, your W3 & Q2 experiment results). Second, We conduct additional comparative experiments involving two alternative splitting strategies that were explored but ultimately not adopted during this research. One is simple sentence splitting, and the other is progressive semantic primitive extraction. Specifically, the simple sentence splitting employs LLM to split all complex sentences in the caption into simple sentences. The progressive semantic primitive extraction employs LLM to extract all objects, relations, and attributes, then reconstructs the original caption into a set of three progressive sentence collections following O∪O-R∪O-R-A. The comparative results against our final adopted  strategy are as follows:
>
> |Method|CLIPScore↑|FID↓|
> |-|-|-|
> |Original Caption |31.46|24.65|
> |Ours (i.e. Hierarchical Input)|32.33|24.18|
> |Simple Sentence Splitting|32.18|24.41|
> |Progressive Primitive Extraction |31.74|24.56|
>
> Experiment setting: 50K dataset, 5 epoch, base model: DiT-ST Medium. As demonstrated, our adopted splitting strategy with hierarchical input achieves the best performance. Regarding your concern in the weakness section about whether our proposed strategy is the best one. We firmly believe that more effective splitting strategies will be developed in the future. However, among all the strategies we have explored thus far, our splitting strategy delivers the best results. If you have better ideas, we would be delighted to engage in further discussion with you, and even potentially collaborate on future research in this area.
>
> &nbsp;
>
> >**W3 & Q2: Results without model fine-tuning**
>
> This is an interesting question. Building upon Table 5, we supplemented our evaluation with experiments using models without any fine-tuning. Specifically, we compare: vanilla SDv3 Medium, SDv3 Medium fine-tuned on our 100K dataset with original captions for 20 epochs (corresponding to SDv3 Medium in Table 5), and DiT-ST Medium trained on our 100K dataset using our method (corresponding to DiT-ST Medium without primitive injection in Table 5). The comparative results for both original caption and split caption are presented below.
>
> |Method|CLIPScore↑|FID↓|
> |-|-|-|
> |Vanilla SDv3+ Original Caption|30.84|24.77|
> |Vanilla SDv3+ Split Caption|31.52|24.52|
> |Finetuning SDv3+ Original Caption|31.31|24.66|
> |Finetuning SDv3+ Split Caption|32.42|24.05|
> |DiT-ST+ Original Caption|31.68|24.61|
> |DiT-ST+ Split Caption|32.81|23.85|
>
> We found that compared to original captions, split captions can achieve better performance even in vanilla SD without any fine-tuning, but fine-tuned models provide greater performance gains.
>
> &nbsp;
>
> >**Q3: Ablation study for Text Encoding Refinement**
>
> Thank you for pointing out this issue, and we sincerely apologize for the absence of this necessary ablation study. During our investigation, we discovered that when utilizing MM-DiT's encoder, the concatenation of two token sequences results in partial dimension capacity remaining underutilized. Therefore, we incorporated complete-text captions in this component, which is our proposed Text Encoding Refinement. Consequently, we conduct ablation experiment on Text Encoding Refinement with DiT-ST Medium. (50K dataset, 5 epochs, base model: DiT-ST Medium)
>
> |Method|CLIPScore↑|FID↓|
> |-|-|-|
> |w/ Text Encoding Refinement|32.33|24.18|
> |w/o Text Encoding Refinement |32.13|24.31|
>
> As shown in the table, without Text Encoding Refinement, the model performance has declines of 0.62% in CLIPScore and 0.54% in FID metrics. Although the improvement from adding complete-text captions is modest, this demonstrates that Text Encoding Refinement indeed contributes positively to model performance. We commit to incorporating this ablation study into the final version. Thank you once again for helping make our paper more complete and solid!
>
> &nbsp;
>
> >**Q4: Effect on downstream tasks**
>
> Thank you for raising this question. We agree that assessing a method’s generality and applicability to downstream tasks is important. Our approach is well-suited for such tasks—including editing—due to its lightweight and modular design. Specifically, split-text conditioning does not alter the pretrained diffusion backbone, but instead redesigns the conditioning mechanism by injecting semantic components (object, attribute, relation) incrementally during denoising, rather than feeding the full prompt at the initial step. This progressive injection aligns better with the evolving nature of generation and ensures compatibility with downstream pipelines.
>
> Taking image editing as an example, which often requires localized modifications (e.g., changing object color or inserting attributes), our method provides two main benefits. First, semantic disentanglement during training allows fine-tuning to target specific elements without affecting unrelated regions, enabling precise compositional edits. Second, staged injection introduces temporal disentanglement, permitting edits at the right step (e.g., modifying attributes without disturbing layout), thus improving controllability. As an architecture-agnostic, plug-and-play module, our approach naturally complements methods like LoRA or T2I-Adapter for editing tasks.
>
> &nbsp;
>
> >**L1: Scaling law for dataset size**
>
> Thank you for pointing out this overlooked issue. We design two related experiments to investigate the scaling law regarding data scale. We use DiT-ST Medium as the base model and, using the same method employed to create our 100K dataset, generate additional 50K and 500K datasets. We trained the base model on datasets of 50K, 100K, and 500K scales for 5 epochs each, then conduct comparative analysis.
>
> |Dataset|CLIPScore↑|FID↓|
> |-|-|-|
> |50K|32.33|24.18|
> |100K|32.71|23.92|
> |500K|33.84|23.40|
>
> Both metrics show clear benefits from data scaling: CLIPScore increases and FID reduces, indicating gains in both semantic alignment and visual quality as training data grows.
>
> |Dataset|[10,15]|[15,25)|[25,35)|[35,45)|[45,55]|Average CLIPScore|
> |-|-|-|-|-|-|-|
> |50K|31.84|32.46|33.40|31.63|32.32|32.33|
> |100K|31.87|32.53|33.60|32.42|33.12|32.71|
> |500K|32.36|33.01|34.04|34.45|35.68|33.84|
>
> By comparing the CLIPScore performance of models trained on different data scale across various caption lengths, we observe that larger training datasets lead to better overall performance. Moreover, models trained on larger datasets demonstrate particular improvements on longer and more complex captions.
>
> To be honest, these two experiments are insufficient to fully demonstrate the model's scaling law with respect to data scale. However, due to resource and funding limitations, we are unable to create larger datasets for further exploration. Therefore, we believe our method **preliminarily** follows the scaling law regarding dataset. We will incorporate this limitation into the final version with more detailed analysis. Thank you for the inspiration!
>
> &nbsp;
>
> >**L2: Limitation of universality**
>
> We appreciate this important reminder. Although our discussion in W1 demonstrates that our method can be transferred to Flux and HunyuanDiT, we honestly acknowledge that our method may not be applicable to all diffusion models—a common limitation shared by similar works. Therefore, we decide to add this limitation discussion to the limitation section in the final version to enhance the depth of our research and analysis. Thank you once again for your constructive and insightful feedback.
>
> &nbsp;
>
> &nbsp;
>
> Once again, we sincerely thank you for dedicating your valuable time to review our submission and for providing so many constructive comments. We are committed to incorporating the analyses and experiments you recommended into the final version, as a way to truly honor your time and effort. To be honest, you are a pivotal reviewer whose feedback could be decisive for the outcome of this paper. We sincerely hope to earn your encouragement and support. Due to the character limit, we can only provide brief responses at this stage. We very much look forward to engaging in further discussion with you during the discussion period. If you have any additional questions or find any of our explanations unclear, please don't hesitate to let us know—we would be happy to provide further clarifications. Thank you once again!
>
> &nbsp;
>
> Best wishes and regards,
>
> All authors of Submission 510

---

### Official Review · Reviewer_1VDF · 2025-07-04

**Clarity:** 3
**Significance:** 2
**Originality:** 2
**Rating:** 3
**Confidence:** 2

**Summary:**

Summary
This paper proposes DiT-ST, a novel split-text conditioning framework for text-to-image diffusion models. Instead of injecting the full caption all at once—which can be semantically dense and challenging for the model to parse—DiT-ST decomposes the caption into multiple semantic primitives (e.g., objects, attributes, relations) using a large language model. These primitives are then hierarchically sorted and injected at different denoising timesteps via cross-attention during the generation process. The motivation is that the progressive nature of diffusion is better suited for staged semantic guidance. Experiments show consistent improvements over strong baselines, demonstrating improved generation quality and semantic alignment

**Questions:**

Please refer to the weakness section.

**Ethical Concerns:**

["NO or VERY MINOR ethics concerns only"]

**Final Justification:**

After reading the SCoPE paper mentioned by another reviewer, I believe the novelty of this work is indeed weakened, though I still find some merits in the work, so I will change my recommendation to borderline reject.

**Limitations:**

yes

**Quality:**

3

**Strengths And Weaknesses:**

Strengths

• The paper is clearly written, and the core idea—decomposing long captions and aligning semantic primitives with different timesteps—is intuitive and elegant.

• The proposed semantic scheduling strategy represents a meaningful extension of earlier works that perform conditioning at fixed stages. By tailoring the conditioning granularity and injection timing, DiT-ST introduces a more structured and interpretable control mechanism.

• The method is modular, requires no changes to the underlying diffusion backbone, and can be readily adapted to existing models.

• The experiments are comprehensive, spanning both qualitative visualizations and quantitative results on multiple datasets and text complexity levels. The SNR-based analysis is particularly insightful and well-motivated.

• The paper includes ablations on primitive types, text parsing, and injection schedules, helping the reader understand where the gains come from

Weaknesses

• The formulation in Equation (10) lacks clarity—particularly, the attention loss term L_{\text{attn}} is not defined in full. It’s unclear what form it takes, what layers it applies to, and how it affects optimization. This should be made explicit for reproducibility.

• The choice of injection timesteps for different semantic primitives (e.g., attributes) is motivated by the convergence of cross-attention outputs, but this assumption is not backed by empirical visualizations. As with Figure 5 (which illustrates the SNR inflation point), the paper would benefit from an analogous figure showing how cross-attention actually converges over time.

---

> ### Author Rebuttal · Authors · 2025-07-29
>
> Hi, Reviewer 1VDF. We sincerely appreciate you taking time for review and providing encouraging and constructive feedback! We value each of them and provide responses below:
>
> >**W1: More detailed elaboration on Equation 10**
>
> We thank the reviewer for the question regarding the design of our attention loss. Our proposed loss term $L_{attn}$  is specifically designed to regulate cross-attention behavior in the multi-stage semantic injection process of diffusion models, ensuring stable training, semantic alignment, and independence between injection stages. The loss is formulated as:
>
> $L_{attn} = \alpha \cdot L_{inject} + \beta \cdot L_{conv} + \eta \cdot L_{mutex}$
>
> where $\alpha = 0.6, \ \beta = 0.25, \ \eta = 0.15$ , and a global scaling factor $\lambda = 0.25$ are used in practice.
>
> The first term, $L_{inject} = E_{t,x_0} \left[ \sum_{i=1}^3 \delta_i(t) \, ( CrossAttn_i(c_{base}, c_{inject}^{(i)}) - c_{target}^{(i)} )^2 \right]$ , enforces the alignment of cross-attention outputs with the target semantics at each activated injection stage, thereby improving semantic precision.
>
> The second term, $L_{conv} = E_t \Big[ \sum_{i=1}^3 \big( SNR_{attn}^{(i)}(t) - SNR_{target}^{(i)}(t) \big)^2 \Big]$, supervises the evolution of the signal-to-noise ratio (SNR) of attention across timesteps to ensure timely convergence, avoiding premature or delayed semantic injection that could otherwise lead to generation bias.
>
> The third term, $L_{mutex} = E_t \Big[ \sum_{i \neq j} \delta_i(t) \cdot \delta_j(t) \cdot \text{Overlap}(\text{AttnMap}_i, \text{AttnMap}_j) \Big]$ , prevents spatial overlap between attention maps of different stages, thereby reducing semantic interference and preserving the independence and interpretability of each injection phase.
>
> Together, these components form a principled objective that not only stabilizes training but also enhances fine-grained semantic alignment, which is critical for the effectiveness of our split-text conditioning strategy in complex prompt scenarios.
>
> &nbsp;
>
>
> >**W2: Visualization of attention at various injection timesteps**
>
> First, we sincerely apologize that due to NeurIPS 2025 policy restrictions, we are prohibited from using or updating any links in the rebuttal, so we cannot show you visualized images. We appreciate you pointing out this issue, which is very necessary. A comprehensive analysis needs to include failure cases.
>
> To determine the attribute injection timestep $T_{\mathrm{inject}}$, we analyze cross-attention maps during denoising. For each candidate $T_{\mathrm{inject}}$, we inspect both the map at this timestep and the following 25 noise level $(T_{\mathrm{inject}}, T_{\mathrm{inject}}+25)$ to verify convergence. The cross-attention map is defined as:
>
> $A_t = \frac{1}{M \cdot H} \sum_{m=1}^M \sum_{h=1}^H \mathrm{Softmax}\left(\frac{Q_t^{(m,h)} K_t^{(m,h)T}}{\sqrt{d_h}}\right)$,
>
> where we further compute entropy $H(A_t)$ for dispersion and contrast ratio $C(A_t) = \frac{\max(A_t)}{\mathrm{mean}(A_t)}$ for focus.
>
> At $T_{\mathrm{inject}}$, entropy drops and contrast rises sharply, indicating a shift from diffuse global attention to localized attribute focus. Over the next 25 noise levels, these metrics stabilize, confirming effective attribute-level conditioning. Furthermore, we compare attention maps at three candidate ranges: 350–375, 400–425, and 450–475.
>
> In the first range (350–375), the model remains in a global layout construction phase, with attention maps characterized by **broad, low-intensity coverage** lacking attribute-specific focus. In contrast, in the last range (450–475), the model has already transitioned into an overly localized state, where attention maps exhibit **prematurely concentrated patterns with reduced semantic plasticity**, indicating that attribute injection would come too late. Only the 400–425 range displays a clear transition from global to localized attribute-focused attention, showing **sharp, high-contrast regions** precisely aligned with attribute semantics. Based on this observation, we select 400 as the attribute injection step in the main text.
>
> These results validate our injection-step selection and provide interpretable evidence of stage-wise semantic alignment.
>
> We sincerely thanks for your valuable suggestions and agree on the importance of this analysis. We apologize once again for not being able to include direct visualizations in the rebuttal due to NeurIPS 2025 policy restrictions. We commit to providing comprehensive visualizations and detailed analyses in the camera-ready version to address this point thoroughly.
>
>
>
> &nbsp;
>
> &nbsp;
>
> Once again, we sincerely thank you for dedicating your valuable time to review our submission and for providing such encouraging feedback. We are committed to incorporating the analyses and experiments you recommended into the final version, as a way to truly honor your time and effort. Although the rebuttal policy limits our ability to provide relevant visualizations that could further enhance the completeness and interpretability of our work, we genuinely hope to receive your continued support in the subsequent stages, which will be crucial to the success of this submission. We look forward to further discussions with you during the discussion period. Thank you again and we eagerly await your response.
>
> &nbsp;
>
> Best wishes and regards,
>
> All authors of Submission 510

---

> > ### Comment · Area_Chair_HjGx · 2025-08-05
> >
> > Dear reviewer 1VDF,
> >
> > We notice that you did not post your response to authors' rebuttal yet. We kindly ask you to read the rebuttals carefully and join the author-reviewer discussions. Your response is critical to ensuring a fair and constructive decision-making process.
> >
> > Best,
> > AC

---

> > ### Comment · Reviewer_1VDF · 2025-08-09
> >
> > The authors’ response satisfactorily addresses my two concerns; I will keep my rating unchanged.

---

> ### Author Response · Authors · 2025-08-07
>
> Hi, Reviewer 1VDF!
>
> We would like to express our heartfelt thanks for you taking your valuable time to review and providing such inspiring feedback. Your suggestions regarding formula clarity and convergence visualization are indeed very helpful for improving the quality of our submission. Following your suggestions, we have made careful revisions in response. We are unable to provide figures for your reference due to NeurIPS policy, and we have chosen to describe our approach details in text. We sincerely hope our revisions meet your expectations. We greatly appreciate your positive affirmation and respectfully hope to gain your continued support! If there are any questions, please don't hesitate to let us know - we would be very happy to respond!
>
> Thank you once again for your time, dedication and support!
>
> &nbsp;
>
> Best wishes and regards,
>
> All authors of Submission 510

---

> > ### Author Response · Authors · 2025-08-09
> >
> > We are glad to learn that our response has addressed your concerns. We would like to express our heartfelt gratitude for taking the time to complete Mandatory Acknowledgement, and for your trust and support in our work. We deeply appreciate all that you’ve done.
> >
> > &nbsp;
> >
> > Best wishes and regards,
> >
> > All authors of Submission 510

---

### Official Review · Reviewer_Dd4v · 2025-07-08

**Clarity:** 4
**Significance:** 3
**Originality:** 3
**Rating:** 5
**Confidence:** 4

**Summary:**

Text-to-image diffusion transformers are typically conditioned on the entire caption during inference. When the caption is long or describes complex relationships between objects, the model struggles to understand the full input. This is due to how tokens compete with each other and how they are positioned in the sentence.
To address this, the paper proposes a method that splits the caption into a hierarchical structure based on its semantic components and their relationships. It has been shown that the convergence of cross-attention in diffusion transformers indicates when semantic objects are formed and when details are refined during inference. By analyzing this convergence, the proposed method decides when each part of the hierarchical structure should be given as condition to the model during inference.
Experiments show that the method performs better on the COCO-5K benchmark and achieves results comparable to state-of-the-art methods on the GenEval benchmark.

**Questions:**

Given that the caption is divided hierarchically, we would expect each text input to the model to be shorter. Since the method uses a pretrained encoder (three encoders, in fact), part of the encoder input might not be used. The paper mentions that the full caption is still provided as input to fully use the encoder. Does this actually improve model performance?

Are there any changes in memory or computational cost when using the proposed method?

Could the method also be applied during training? I'm not asking the authors to implement this, but I would appreciate a discussion of the potential challenges and benefits.

**Ethical Concerns:**

["NO or VERY MINOR ethics concerns only"]

**Final Justification:**

The authors addressed my questions. I have read the other reviews and replies and I can confidently recommend this paper to be accepted.

**Limitations:**

Limitations are discussed in the appendix.

**Paper Formatting Concerns:**

No concerns

**Quality:**

3

**Strengths And Weaknesses:**

The paper is clearly written and easy to follow. The proposed method is simple to implement and can be applied to multiple models at inference time.

The idea of conditioning the model on different semantic parts of the caption at different inference steps is original, and using cross-attention convergence to guide this process seems sound.

The experimental results are convincing, and the ablation studies show that the method’s components contribute effectively.

Overall, this is a solid paper. I look forward to seeing the other reviews and the authors’ responses.

---

> ### Author Rebuttal · Authors · 2025-07-29
>
> Hi, Reviewer Dd4v. We sincerely appreciate you taking time for review and providing   encouraging and constructive feedback! We value each of them and provide responses below:
>
> >**Q1: Ablation study for Text Encoding Refinement**
>
> Thank you for pointing out this issue, and we sincerely apologize for the absence of this necessary ablation study. During our investigation, we discovered that when utilizing MM-DiT's encoder, the concatenation of two token sequences results in partial dimension capacity remaining underutilized. Therefore, we incorporated complete-text captions in this component, which is our proposed Text Encoding Refinement. Consequently, we conduct ablation experiment on Text Encoding Refinement with DiT-ST Medium. (50K dataset, 5 epochs, base model: DiT-ST Medium)
>
> |Method|CLIPScore↑|FID↓|
> |-|-|-|
> |w/ Text Encoding Refinement|32.33|24.18|
> |w/o Text Encoding Refinement |32.13|24.31|
>
> As shown in the table, without Text Encoding Refinement, the model performance has declines of 0.62% in CLIPScore and 0.54% in FID metrics. Although the improvement from adding complete-text captions is modest, this demonstrates that Text Encoding Refinement indeed contributes positively to model performance. We commit to incorporating this ablation study into the final version. Thank you once again for helping make our paper more complete and solid!
>
> &nbsp;
>
> >**Q2: Analysis of memory usage and computational cost**
>
> First, we analyze the additional memory overhead introduced by Split-Text Conditioning by listing the formula. The additional memory cost mainly arises from text embedding caching and cross-attention operations.
>
> We employ text encoders such as CLIP-L $[B,77,768]$, CLIP-G $[B,77,1280]$ and T5-XXL $[B,256,4096]$ with $B$ for batchsize, $[77,256]$ for the number of token respectively and $[768, 1280,4096]$ for embedding  dimension. Given a full-text length $L$, object, relation, and attribute segments $L_O, L_R, L_A$, embedding dimension $d$ , batch size $B$, and fp16 precision, the memory required for a single text embedding is $M_{\text{text, single}} \approx B \cdot L \cdot d \cdot \mathrm{sizeof(fp)}$ .  After O-R-A segmentation, we cache three segmented embeddings and the full-text embedding, yielding $M_{\text{text, ORA}} \approx 4 \cdot B \cdot L \cdot d \cdot \mathrm{sizeof(fp)}$ . Since $L_O + L_R + L_A \approx L$ , the main increase comes from storing multiple embeddings rather than longer sequences.
>
> For cross-attention, which dominates the additional memory overhead, let the number of image tokens be $N = (imgsize // 2)²=512 \times 512$,  the number of attention heads $H=28$, and the per-head dimension $d_h=64$, giving a total hidden dimension $d=H \times d_h=1792$ . The key-value (KV) cache memory for a single cross-attention layer is $M_{\text{KV}} \approx B \cdot H \cdot N \cdot L_{\text{avg}} \cdot d_h \cdot \mathrm{sizeof(fp)}$ . Since our method injects three cross-attention stages during denoising, the total KV cache memory is $M_{\text{KV, total}} \approx 3 \cdot B \cdot H \cdot N \cdot L_{\text{avg}} \cdot d_h \cdot \mathrm{sizeof(fp)}$ , while the attention logits require an additional $M_{\text{logit}} \approx 3 \cdot B \cdot H \cdot N \cdot L_{\text{avg}} \cdot \mathrm{sizeof(fp)}$ . Therefore, the overall memory cost is $M_{\text{total}} \approx 4 \cdot B \cdot L \cdot d + 3 \cdot B \cdot H \cdot N \cdot L_{\text{avg}} \cdot d_h + 3 \cdot B \cdot H \cdot N \cdot L_{\text{avg}}$ , where the first term corresponds to text embedding caching, the second to KV cache, and the third to attention logits, with cross-attention being the dominant factor while convergence analysis is negligible.
>
> In terms of computational cost, evaluated on an NVIDIA H200 GPU, the overhead of Split-Text Conditioning is modest and results are as follow:
>
> |Strategy|Throughout(img./s)↑|GFLOPs(G/img.)↓|
> |-|-|-|
> |DiT-ST w/ original caption w/o injection (512×512)|0.2262|605|
> |DiT-ST w/ hierarchical input w/o injection (512×512)|0.2248|710(+17.4%)|
> |DiT-ST w/ hierarchical input w/ injection (512×512)|0.2161|727(+20.2%)|
> |DiT-ST w/ original caption w/o injection (1024×1024)|0.0572|7310|
> |DiT-ST w/ hierarchical input w/o injection (1024×1024)|0.0571|7410(+1.4%)|
> |DiT-ST w/ hierarchical input w/ injection (1024×1024)|0.0558|7530(+3%)|
>
> At 512×512, replacing the original caption with hierarchical input increases GFLO
> Ps by 17.4% but has negligible impact on throughput (0.2262 → 0.2248 img/s), as text processing is a minor component of the overall cost. Adding injection further raises GFLOPs by 20.2% and slightly reduces throughput (0.2248 → 0.2161 img/s) due to the additional cross-attention computations.
>
> At 1024×1024, where image generation dominates the cost, hierarchical input adds only 1.4% GFLOPs with no throughput change, while injection incurs an extra 3% GFLOPs with a small throughput drop (0.0571 → 0.0558 img/s).
>
> Overall, text-related overhead becomes negligible at high resolution, and injection introduces modest, manageable additional cost.
>
> &nbsp;
>
> >**Q3: Discussion of challenges and benefits during training**
>
> Thank you for your question. Our DiT-ST model **already incorporates staged semantic injection during training**, and this design has provided clear empirical benefits. First, training-time injection explicitly guides the model to learn the temporal scheduling of object, relation, and attribute conditioning, leading to more robust semantic disentanglement compared to inference-only strategies. Second, it improves generalization and controllability: by integrating injection into the optimization process, the model not only performs well within the training distribution but also exhibits stronger interpretability and stability for long prompts and complex scenes. Third, training-time injection stabilizes the behavior of cross-attention modules, reducing the need for additional heuristic scheduling at inference time.
>
>  However, we also note that this approach introduces additional costs, including higher memory consumption and longer training time due to multi-stage injection and the associated cross-attention parameters and KV caches. Moreover, the scheduling hyperparameters for injection require careful tuning during training.
>
> For comparison, ControlNet, a well-known approach that **also integrates conditioning during training**, demonstrates similar benefits: it enables the model to directly learn the correspondence between control signals (e.g., edges, depth, pose) and image generation, yielding strong controllability at inference. Yet, ControlNet also faces limitations, such as increased parameter count from additional conditional branches and the need for specialized annotated data, which reduces its flexibility.
>
> In contrast, SCoPE performs **interpolation only at inference time**, which avoids the training overhead but fails to expose the model to semantic injection during optimization. As a result, SCoPE does not fully resolve semantic entanglement and lacks explicit modeling of different semantic types (objects, relations, and attributes). This inference-only design can lead to less stable performance, especially for long prompts or compositionally complex scenes.
>
> In summary, incorporating injection during training—as we do in DiT-ST—offers clear advantages for semantic disentanglement, interpretability, and controllability, distinguishing it from inference-only approaches such as SCoPE.
>
>
> &nbsp;
>
> &nbsp;
>
> Once again, we sincerely thank you for dedicating your valuable time to review our submission and providing such positive feedback. We are committed to incorporating the above analyses and experiments you recommended into the final version, as a way to truly honor your time and effort. Your encouragement is an important reason why we are willing to continue participating in the rebuttal process. We eagerly look forward to your continued support, which will play a crucial role in the success of this submission. Therefore, we very much anticipate further discussion with you during the discussion period. If you have any other questions, please kindly let us know, and we will be happy to response them enthusiastically. Thank you!
>
> &nbsp;
>
> Best wishes and regards,
>
> All authors of Submission 510

---

> > ### Comment · Reviewer_Dd4v · 2025-08-08
> > **Official comment**
> >
> > I thank the author for the thorough reply! I think this is a solid paper and I maintain my positive score.

---

> ### Author Response · Authors · 2025-08-07
>
> Hi, Reviewer Dd4v!
>
> We would like to sincerely thank you for taking the time to review our paper as an emergency reviewer and for giving us such generous encouragement, which is extremely precious to us. At the same time, your constructive comments have also given us much valuable inspiration, and we have responded to each one carefully in our rebuttal, hoping that you will be satisfied. Your encouragement has always been a constant source of motivation for us to improve the quality of our research, and we deeply value your continued support. If there is anything that needs additional explanation or supplementation, we would be more than happy to do so!
>
> Thank you once again for your time, dedication and support!
>
> &nbsp;
>
> Best wishes and regards,
>
> All authors of Submission 510

---

> > ### Author Response · Authors · 2025-08-08
> >
> > Thank you for having completed the Mandatory Acknowledgement. We deeply appreciate all that you’ve done.
> >
> > &nbsp;
> >
> > Best wishes and regards,
> >
> > All authors of Submission 510

---

### Official Review · Reviewer_6Z8X · 2025-07-08

**Clarity:** 3
**Significance:** 2
**Originality:** 2
**Rating:** 3
**Confidence:** 3

**Summary:**

This paper introduces a new framework DiT-ST which aims to handle the inherent limitations of existing text-to-image models. Specifically,  understanding complex long texts through a technique called split-text conditioning is not trivial.
This framework leverages a large language model to parse a long textual description into a set of simpler, hierarchical short phrases.
These phrases are then incrementally injected into different stages of the denoising process.
The experiments show that it significantly improves the generated image's detail accuracy and semantic consistency and the method outperforms or rivals state-of-the-art models with minimal parameter increasement.

**Questions:**

The authors are encouraged to discuss the differences to the refered paper and complement necessary experiments.

**Ethical Concerns:**

["NO or VERY MINOR ethics concerns only"]

**Final Justification:**

Thank the authors for partially addressin the concerns and the reviewer decided to raise the score to 3.

**Limitations:**

Yes.

**Paper Formatting Concerns:**

No.

**Quality:**

2

**Strengths And Weaknesses:**

Strengths:
- This paper uses a simple appraoch to let the model more accurately understand and generate fine-grained details from the input. The method is splitting complex text into simple subject-relation-attribute meta-phrases.
- The method is parameter-efficient and architecture-agnostic as the core idea lies in preprocessing the input text and injecting it in stages without requiring major modifications to the model architecture.
- Experiments show that this method performs well in very long prompts with satisfying CLIPScore performance. Also on the comprehensive benchmark GenEval the DiT-ST model outperforms the SDv3 Medium baseline in many dimensions.

Weaknesses:

- This paper lacks an important reference of "Progressive Prompt Detailing for Improved Alignment in Text-to-Image
Generative Models" (https://arxiv.org/pdf/2503.17794v1) which has a very similar method on the same issue. This papper should discuss the differences.
- The evaluation is not enough. The evaluation metrics are not comprehensive. Though FID is adopted to evaluate the image quality while it is limited. Also, this paper only adopts GenEval and CLIPScore to test the text-image alignment without a user study.
- The experiments are not comprehensive. Only one LLM is adopted for the experiments. Other LLMs should also be tested for the effectiveness and generalization. Also, this paper does not discuss if finetuning LLM works better for this task.
- The experiment does not include failure cases and analyze them.
- The captions should be more detailed to better illustrate the content and the motivation of the figures and tables.

---

> ### Author Rebuttal · Authors · 2025-07-28
>
> Hi, Reviewer 6Z8X. We sincerely appreciate your careful review and so many constructive comments. We value each of them and provide responses below:
>
> >**W1: Difference between our work and SCoPE**
>
> Thank you for the vital reference! We were already aware of this work and cited it in Line 104. We fully agree with your suggestion and summarize detailed differences in **method** and **experiment**:
>
> **+Method**
>
> ① Method process.
>
> Both DiT-ST (Ours) and SCoPE aim to mitigate semantic overload through progressive injection, thereby enhancing the model's understanding of captions and improving generation quality, However, the two methods differ fundamentally in design.
>
> SCoPE employs LLM to segment captions into multiple sub-captions progressing from coarse to fine-grained. The timestep for progressive injection is determined by calculating similarity among sub-captions and applying proportional similarity ratios. While SCoPE achieves caption segmentation through this training-free strategy, it cannot distinguish semantic primitives (object, relation, attribute), thereby remaining vulnerable to semantic competition and concept coupling problems outlined in our introduction.
>
> We employ LLM to split captions into three semantic primitives—objects, relations, and attributes—constructing independent encoding streams. By utilizing SNR curvature and cross-attention convergence, we can easily identify optimal injection timesteps, enabling the model to first establish object concepts before incrementally incorporating relation and attribute. Our method effectively mitigates semantic competition and significantly enhances both generation quality and semantic consistency.
>
> Therefore, SCoPE tends to focus on coarse-grained input control during the inference stage, whereas ours represents a fine-grained structural optimization method targeting internal model architecture and modeling processes. DiT-ST demonstrates superior interpretability and controllability.
>
> ② Model structure.
>
> SCoPE, as a method based on the U-Net architecture, has a multi-scale convolutional design that is restricted by a local receptive field, which makes it difficult to adequately capture long-distance dependencies and has limited scalability. SCoPE tends to be a heuristic inference enhancement rather than a scalable training framework.
>
> Ours is based on the Diffusion Transformer, which is naturally adapted to larger scales and more complex scenarios. Global self-attention mechanism has capabilities for long-distance semantic modeling and complex scenario understanding, and makes our method preliminarily follows the scaling law.
>
> **+Experiment**
>
> ① Evaluation metrics.
>
> SCoPE uses CLIPScore and VQAScore.
>
> We use FID, CLIPScore, GenEval and VQAScore (Inspired by the reference you provided).
>
> ② Base models.
>
> SCoPE uses U-Net-based diffusions: SD 1–4, SD 2.1, and SDXL.
>
> We use Transformer-based diffusions: SDv3 Medium and SDv3.5 Large. Thus, our method preliminarily follows the scaling law (Table7, `Reviewer kbep` L1 and `Reviewer kbep` W3 & Q2).
>
> ③ Experiment evaluation.
>
> SCoPE only evaluates VQAScore and CLIPScore for proposed methods and base models, without more main results or any ablation studies.
>
> Our main experiments evaluate FID, CLIPScore, GenEval, and VQAScore (added in W2 based on your suggestion), along with model performance under different caption lengths and model sizes. We include not only base models (SD) but also Transformer-based diffusions such as PixArt-α, DALL-E 3, Flux.1 Dev, Dream Engine. In addition, extensive ablation studies further validate our design's effectiveness.
>
> ④ Method performance.
>
> SCoPE improves CLIPScore by up to 1.3% over the base model (SDv2.1).
>
> Ours improve CLIPScore by up to 2.78% over the base model (SDv3 Medium).
>
> As two methods aren’t directly comparable, we use CLIPScore gain as a rough reference. Notably: 1.SCoPE is training-free, which is its advantage. 2.SCoPE adopts prompt enhancement, it’s unclear how much of the gain comes from their design instead of the added information. 3.SCoPE generates eight candidate outputs per prompt and selects the best-aligned one, while we use a single random one.
>
> ---
>
> In addition, our models and datasets are now publicly available. Compared to 4-page reference, our paper offers greater detail, completeness and more visualizations. Generally, SCoPE is still an interesting work. We will incorporate above analysis into the final version to strengthen our study.  Sincerely thank you for your insightful input!
>
> &nbsp;
>
> >**W2: More comprehensive evaluation**
>
> We fully agree with you. CLIPScore and FID have inherent limitations, as they cannot adequately evaluate image naturalness, fidelity, and rationality. Therefore, we additionally adopted GenEval for more comprehensive evaluation, such as semantic consistency, fidelity, clarity, diversity, and plausibility. While many other metrics exist, these three are the most widely adopted in the field. Unfortunately, when seeking to conduct broad comparative analysis, the number of applicable universal metrics becomes limited. Fortunately, VQAScore from the reference you provided is also a general benchmark similar to GenEval. In order to reserve enough space for responding to your other valuable feedback, we kindly direct you to Reviewer `4VWJ` W4 for our VQAScore evaluation. Thank you for the inspiration!
>
> As for the user study, we acknowledge this limitation in the appendix (lines 969–973). As it's a common limitation in all similar works [9][10][22][26][39][45] (same reference in our submission) [r1][r2][r3][r4], we regret we couldn't conduct one. As an alternative, we offer extensive visualizations and available models to facilitate your subjective perception and evaluation. We apologize once again for our inability and sincerely appreciate your understanding.
>
> [r1] Reconstruction vs. Generation: Taming Optimization Dilemma in Latent Diffusion Models, CVPR 2025
>
> [r2] Language-Guided Image Tokenization for Generation, CVPR 2025
>
> [r3] SANA: Efficient High-Resolution Text-to-Image Synthesis with Linear Diffusion Transformers, ICLR 2025
>
> [r4] SinSR: Diffusion-Based Image Super-Resolution in a Single Step, CVPR 2024
>
> &nbsp;
>
> > **W3: LLM-related experiments**
>
> Thank you for pointing out our overlooked experiments. We add 2 experiments:
>
> ① Different LLMs. Besides initial Qwen-plus, we also use GPT-3.5-turbo and Gemini-2.0-flash and evaluate them on COCO-5K. (50K dataset, 5 epochs, base model: SDv3 Medium)
>
> |LLM|CLIPScore↑|FID↓|
> |-|-|-|
> |w/o (Original caption)|31.05|24.70|
> |Qwen|32.33|24.18|
> |GPT|32.28|24.15|
> |Gemini|32.22|24.23|
>
> We observe that using any LLM improves performance, with Qwen performing best. This demonstrates the effectiveness and generalization of LLMs in our method.
>
> ② LLM finetuning: Prompting is viewed as LLM finetuning. Besides our prompt, we also compare a simple prompt and w/o prompt on COCO-5K. The simple prompt is: *Extract all objects, relation between object and attributes from the description.* (50K dataset, 5 epochs, base model: SDv3 Medium)
>
> |Method|CLIPScore↑|FID↓|
> |-|-|-|
> |Our prompt|32.33|24.18|
> |Simple prompt|32.18|24.41|
> |w/o prompt|30.45|24.52|
>
> It shows that prompt-based LLM finetuning improves performance, with better prompts yielding better results. Without prompting, LLM tends to generate low-quality expansions [r5], hindering semantic understanding of captions.
>
> [r5] Exploring the Role of Large Language Models in Prompt Encoding for Diffusion Models, NeurIPS 2024
>
> &nbsp;
>
> > **W4: Failure case analysis**
>
> First, we sincerely apologize that due to NeurIPS 2025 policy restrictions, we are prohibited from using or updating any links in the rebuttal, so we cannot show you visualized images. We appreciate you pointing out this issue, which is very necessary. A comprehensive analysis needs to include failure cases.
>
> +**Prompt of High Complexity**
>
> The first case uses a prompt with complex syntax and layered semantics, *describing a studio scene with ivy-covered windows, a weathered violin, wine glass, lace-covered table, and and a hidden cat watching dust*. Both our model and SDv3 failed to fully capture some attributes (e.g., the violin’s aged texture), likely due to implicit and long-range phrasing. Still, our model better preserves spatial cues such as the cat’s occlusion and gaze, often missed by baselines. While imperfect, our method generally retains more semantic detail in such complex prompts.
>
> +**Prompt of Irrelevant Information**
>
> The second case shows failure due to missing visual grounding in the input. The prompt *You may come for your appointment at 9 a.m. tomorrow.* contains no visual object or scene. Although users may expect a hospital-related image, all models—including ours—produce irrelevant outputs. This reflects a limitation of T2I models when prompts lack explicit semantics.
>
> We hope these cases help clarify the scope and limitations of our method under different prompt conditions.
>
> &nbsp;
>
> > **W5: More detailed captions**
>
> Thanks for your suggestion! We promise to detail captions of tables and figures to better illustrate the content and the motivation.
>
> &nbsp;
>
> &nbsp;
>
> Once again, we sincerely thank you for dedicating your valuable time to review our submission and offering so many insightful and constructive suggestions. We are committed to incorporating the above analyses and experiments you recommended into the final version, as a way to truly honor your time and effort. Honestly, you are the most influential reviewer for the outcome of this submission, and we eagerly look forward to engaging with you further during the discussion period and hope to gain your continued guidance and encouragement. If you have any other questions or feel that any of our explanations are unclear, please kindly let us know—we will be happy to provide further detailed clarifications. Thank you very much once again!
>
> &nbsp;
>
> Best wishes and regards,
>
> All authors of Submission 510

---

> ### Author Response · Authors · 2025-08-07
>
> Hi, Reviewer 6Z8X!
>
> We would like to express our heartfelt thanks for you taking the time to review our paper as an emergency reviewer and providing many constructive comments. We value each of your comments and have responded to them one by one in our rebuttal. With less than 3 days remaining until the discussion period ends, we sincerely hope our responses and additional experiments have addressed your concerns. Additionally, we would be deeply grateful for your favorable consideration. As the most important reviewer, any additional support from you could have a better decisive impact. It would not only improve our work, but also reflect the significance of your insightful suggestions. If there’s anything else we can clarify or supplement, please don’t hesitate to let us know, we're ready to respond immediately.
>
> Thank you once again for your time, dedication and support!
>
> &nbsp;
>
> Best wishes and regards,
>
> All authors of Submission 510

---

> > ### Comment · Reviewer_6Z8X · 2025-08-08
> >
> > Thanks for the author's effort on the rebuttal and the added experiments. For W1, the author's explanation clarifies differences from SCoPE, addressing part of the reviewer's concern. The overall idea, progressive, staged semantic injection, remains similar in general, which in the reviewer's view weakens the novelty. For W2, the added GenEval and VQAScore partially address evaluation limits, but the reviewer still believe a user study is needed, since metrics can diverge from human perception as it is a generation task. For W3, adding the LLM-choice and prompt-design experiments makes the evaluation more complete; the reviewer kindly recommend to keep these in the next version of the paper. For W4, the reviewer kindly suggests to include the discussed failure cases and an explicit limitations section in the next version. Overall, thank the authors for partially addressed the reviewer's concerns that the reviewer still have reservations but will raise the score to 3.

---

> > ### Author Response · Authors · 2025-08-09
> >
> > We sincerely thank you for your thorough feedback and for giving us further support. It is very nice that you have pointed out in detail which issues remain your concerns. We greatly respect your views on these issues and are willing to do our best to address them in the final version. Once again, we deeply appreciate all that you’ve done.
> >
> > &nbsp;
> >
> > Best wishes and regards,
> >
> > All authors of Submission 510

---

### Decision · Program_Chairs · 2025-09-17

**Decision:**

Accept (poster)

**Comment:**

### Summary of Claims and Findings

The paper claims that standard text-to-image diffusion models struggle with complex and long-form text prompts. To address this, the authors propose a "split-text conditioning" framework that uses a LLM to parse the complex text prompt into a collection of simpler sentences. These primitives are then injected incrementally into different stages of the diffusion model's denoising process. The experimental results show that their method significantly outperforms the strong baselines on several metrics (CLIPScore, FID, GenEval) particularly on complex text prompts.

### Strengths of the Paper
All reviewers agree that this paper is written well, the proposed method is simple and effective, and shows strong empirical results with sufficient ablations.

### Weaknesses of the Paper
The primary weaknesses were pointed out by reviewer 6Z8X.

- Limited novelty due to a concurrent work: Reviewer 6Z8X pointed out a concurrent work, SCoPE (arxiv.org/pdf/2503.17794v1), which shares similar ideas. This led to a debate about the paper's novelty.
- Lack of User Study.

### Reasons for Recommendation
I am recommending this paper for acceptance. The authors provided a strong rebuttal that addressed most of the weaknesses raised by reviewers. While the concerns about novelty are valid, they are outweighed by the paper's technical merits, the strength of the rebuttal, and the official NeurIPS policy on contemporaneous work.

SCoPE should be considered contemporaneous work under the NeurIPS 2025 policy (as it appeared online after the March 1st deadline) and that a submission should not be rejected solely on this basis. More importantly, they provided a detailed, five-point differentiation, highlighting that their method operates at a more granular "semantic primitive" level, while SCoPE operates at a "sub-prompt" level. Their approach for determining injection timesteps is also more deeply tied to the internal dynamics of the diffusion model. AC would strongly suggest the authors to add this discussion to the final paper.

### Summary of Discussion and Rebuttal
The discussion period was productive and crucial for the final decision. The key points were:
- Novelty vs. SCoPE: This was the main topic. Reviewer 6Z8X initially flagged this, and Reviewers 1VDF and 4VWJ later echoed the concern, with Reviewer 1VDF stating it "weakened the novelty." The authors responded by detailing the technical differences and referencing the conference's contemporaneous work policy. I, as the AC, reminded reviewers of this policy to ensure a fair evaluation. While some reviewers remained concerned, Reviewer kbep (who is more senior) agreed it should be treated as concurrent work. My final decision weighs the authors' technical differentiation and the official policy more.
- Evaluation scope: Reviewers 6Z8X and 4VWJ requested more comprehensive evaluation, particularly a user study. The authors acknowledged the lack of a user study as a common limitation in the field and compensated by adding a new VQAScore benchmark analysis to their rebuttal, which was a strong response.
- Generalizability: Reviewer kbep questioned the method's transferability. The authors addressed this directly by running new experiments showing positive results when applying their framework to two other Transformer-based diffusion models.
- Technical Clarity: Reviewer 1VDF requested clarification on a loss function (Equation 10) and visualizations for attention convergence. The authors provided a detailed mathematical breakdown and a clear textual description of the convergence dynamics, which satisfied the reviewer.

Overall, the authors addressed the reviewers' concerns comprehensively with new experiments and detailed explanations. This significantly improved the paper's quality, justifying the decision to accept the paper.